# Single-cell atlas of the first intra-mammalian developmental stage of the human parasite *Schistosoma mansoni*

Carmen Lidia Diaz Soria [1,5], Jayhun Lee [2,3,5], Tracy Chong[2,3], Avril Coghlan [1], Alan Tracey [1], Matthew D. Young[1], Tallulah Andrews [1], Christopher Hall [1], Bee Ling Ng [1], Kate Rawlinson [1], Stephen R. Doyle [1], Steven Leonard[1], Zhigang Lu [1], Hayley M. Bennett [1], Gabriel Rinaldi [1✉], Phillip A. Newmark [2,3,4✉] & Matthew Berriman [1✉]

Over 250 million people suffer from schistosomiasis, a tropical disease caused by parasitic flatworms known as schistosomes. Humans become infected by free-swimming, water-borne larvae, which penetrate the skin. The earliest intra-mammalian stage, called the schistoso-mulum, undergoes a series of developmental transitions. These changes are critical for the parasite to adapt to its new environment as it navigates through host tissues to reach its niche, where it will grow to reproductive maturity. Unravelling the mechanisms that drive intra-mammalian development requires knowledge of the spatial organisation and tran-scriptional dynamics of different cell types that comprise the schistomulum body. To fill these important knowledge gaps, we perform single-cell RNA sequencing on two-day old schis-tosomula of *Schistosoma mansoni*. We identify likely gene expression profiles for muscle, nervous system, tegument, oesophageal gland, parenchymal/primordial gut cells, and stem cells. In addition, we validate cell markers for all these clusters by in situ hybridisation in schistosomula and adult parasites. Taken together, this study provides a comprehensive cell-type atlas for the early intra-mammalian stage of this devastating metazoan parasite.

[1] Wellcome Sanger Institute, Wellcome Genome Campus, Hinxton, Cambridgeshire, UK. [2] Regenerative Biology, Morgridge Institute for Research, Madison, WI, USA. [3] Howard Hughes Medical Institute, University of Wisconsin-Madison, Madison, WI, USA. [4] Department of Integrative Biology, University of Wisconsin-Madison, Madison, WI, USA. [5]These authors contributed equally: Carmen Lidia Diaz Soria, Jayhun Lee. ✉email: gr10@sanger.ac.uk; PNewmark@morgridge.org; mb4@sanger.ac.uk

Schistosomes are parasitic flatworms that cause schistosomiasis, a serious, disabling, and neglected tropical disease (NTD). More than 250 million people require treatment each year, particularly in Africa[1]. The life cycle of this metazoan parasite is complex. A schistosome egg hatches in water to release a free-living, invasive larva that develops into asexually replicating forms within aquatic snails (the intermediate host). From the snail, thousands of cercariae—a second free-living larval form— are released into freshwater to find and invade a mammal (the definitive host). In the mammalian host, the larvae (schistosomula) migrate and develop into distinctive male or female adult worms[2] (Fig. 1a). While the only drug currently available to treat schistosomiasis (praziquantel) works efficiently to kill adult parasites, it is less effective against immature parasites, including schistosomula[3]. Understanding the parasite's biology is a critical step for developing novel strategies to treat and control this NTD.

During invasion, the parasite undergoes a major physiological and morphological transformation from the free-living, highly motile cercariae to the adult parasitic form[4]. Upon penetration, the tail used for swimming is lost. Less than three hours after entering the host, the thick glycocalyx is removed and the tegument remodelled to serve both nutrient-absorption and immune-protection roles[5]. Throughout the rest of the organism's life span in the definitive host, a population of subtegumental progenitor cells continuously replenish the tegument, allowing the parasite to survive for decades[6,7]. The schistosomula make their way into blood or lymphatic vessels and, one week after infection, reach the lung capillaries[8]. The migration through the lung requires coordinated neuromuscular activities, including cycles of muscle elongation and contraction[9], to squeeze through capillaries and reach the general circulation[8]. Over the following weeks, the parasites mature further into sexually reproducing adults. Dramatic changes to the parasite are required that include posterior growth, remodelling of the musculature[10] and nervous system[11,12] as well as the development of the gonads[13] and gut[14]. This extensive tissue development starts in the schistosomula, with stem cells driving these transitions[7,15,16]. However, to decipher cellular and molecular mechanisms underlying schistosomula development, a detailed understanding of the spatial organisation and transcriptional programs of individual cells is needed.

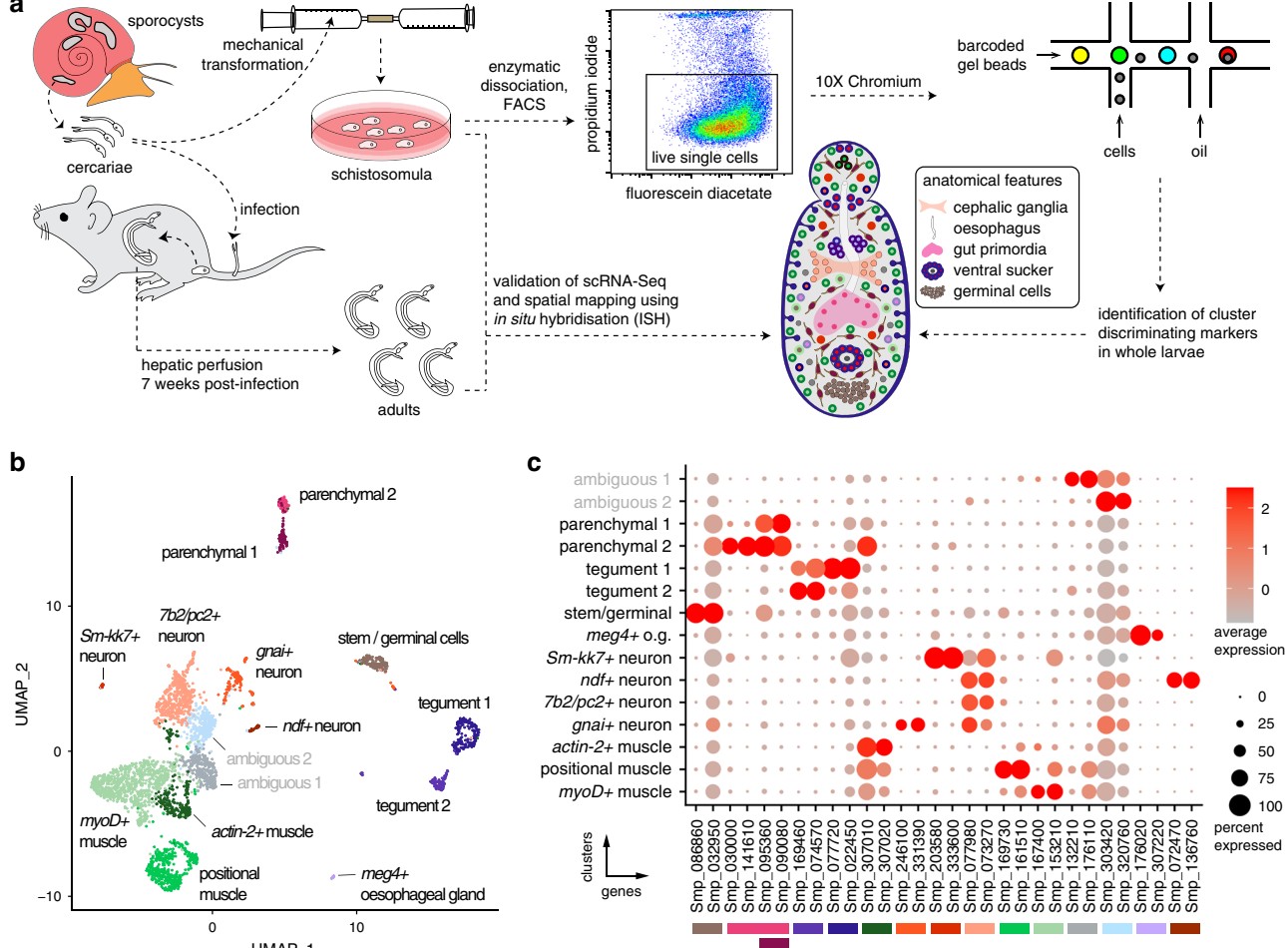

**Fig. 1 Identification of 13 transcriptionally distinct cell types in schistosomula. a** Experimental scheme describing the sources of the parasite material, single-cell analysis and validation pipeline. Approximately 5000 schistosomula per experiment were dissociated, followed by enrichment of fluorescein diacetate (FDA+) live cells using fluorescence-activated cell sorting (FACS). Cells were loaded according to the 10X Chromium single-cell 3′ protocol. Clustering was carried out to identify distinct populations and population-specific markers. Validation of population-specific markers was performed by in situ hybridisation (ISH). **b** Uniform Manifold Approximation and Projection (UMAP) representation of 3226 schistosomulum single cells. Cell clusters are coloured and distinctively labelled. **c** Gene-expression profiles of population markers identified for each of the cell clusters. The colours represent the level of expression from dark red (high expression) to light red (low expression). The sizes of the circles represent the percentages of cells in those clusters that expressed a specific gene. The colour bars under gene IDs represent the clusters in (**b**).

Important insights into major processes that underlie the transformations across the life cycle have been gained from bulk transcriptomic studies[6,7,15–25]. However, these studies are not able to quantify the relative abundance of different cell types from the absolute expression per cell, and the signal from highly expressed genes in a minority of cells can often be masked by a population averaging effect. Single-cell RNA sequencing has previously been used successfully to characterise cell types[26–35] and understand how the cell expression profile changes during differentiation[32–38]. Notable examples include recent studies in the free-living planarian flatworm *Schmidtea mediterranea*[32,33,39], a well-established model for regeneration in the Phylum Platyhelminthes[40].

Here, we have used scRNAseq to characterise two-day schistosomula obtained by in vitro transformation of cercariae[23] using 10X Chromium technology and validated the cell clusters by RNA in situ hybridization (ISH) in schistosomula and adult worms. We identified at least thirteen discrete cell populations, and described and validated novel marker genes for muscles, nervous system, tegument, oesophageal gland, parenchyma/gut primordia and stem cells. This study lays the foundation towards a greater understanding of cell types and tissue differentiation in the first intra-mammalian developmental stage of this NTD pathogen.

## Results

### Identification of 13 transcriptionally distinct cell types in schistosomula.

We performed single-cell RNA sequencing of schistosomula collected two days after mechanically detaching the tail from free-living motile larvae (cercariae) (Fig. 1a). We first developed a protocol to efficiently dissociate the parasites using a protease cocktail, after which individual live cells were collected using fluorescence-activated cell sorting (FACS) (Fig. 1a and Supplementary Fig. 1a). Using the droplet-based 10X Genomics Chromium platform, we generated transcriptome-sequencing data from a total of 3513 larval cells, of which 3226 passed strict quality-control filters, resulting in a median of 900 genes and depth of 283,000 reads per cell and 1268 median UMI counts per cell (Supplementary Data 1). Given that an individual schistosomulum comprises ~900 cells (Supplementary Fig. 1b), the number of quality-controlled cells theoretically represents >2× coverage of all cells in the organism at this developmental stage.

To create a cellular map of the *S. mansoni* schistosomula, we used Seurat[41] to cluster and identify marker genes that were best able to discriminate between populations (Fig. 1b, c and Supplementary Data 2). To identify the cell types that each Seurat cluster represented, we curated lists of previously defined cell-specific markers (Supplementary Data 3). For example, tegument[6,7,42–44] and stem[15,45–47] cell clusters were identified based on known marker genes in *S. mansoni*, whereas muscle cells[48–50] and neurons[22,51–53] were identified based on characterised marker genes in mouse and humans and *S. mansoni* (Supplementary Data 3). Based on the marker genes identified using Seurat, we identified the following distinct clusters of cells: three muscle-like (1,440 cells), two tegumental (281 cells), two parenchymal (158 cells), one cluster resembling stem cells (126), four resembling the nervous system (643 cells), oesophageal gland (17 cells), and two ambiguous clusters (Supplementary Fig. 12) for which no specific markers could be predicted (561 cells). In addition, Gene Ontology (GO) analysis of the marker genes for these two ambiguous clusters did not result in enrichment of any particular processes (Supplementary Fig. 2). Furthermore, the in situ validation of one of the clusters (ambiguous 1) remained inconclusive (Supplementary Fig. 12). For the rest of the populations, the GO analysis generally matched the predicted

cellular processes for each cluster (Supplementary Fig. 2). For instance, as expected, the stem/germinal cell cluster showed a significant enrichment in genes involved in translation. Meanwhile, neuronal cells and muscle cells were enriched in processes involved in GPCR signalling and cytoskeleton, respectively. These analyses suggested that each cluster is molecularly distinct and likely displays different biological functions. Therefore, we defined highly specific cluster-defining transcripts (potential cell markers) (Supplementary Data 2, Supplementary Fig. 3) and characterised their spatial expression in both larval schistosomula and adult schistosomes by ISH (Supplementary Data 4).

### Muscle cells show position-dependent patterns of expression.

Three discrete muscle cell clusters were identified by examining the expression of the well-described muscle-specific genes myosin[54] and troponin[50] (Fig. 2a and Supplementary Fig. 3a), as well as a number of differentially expressed markers. One muscle cluster (428 cells) was distinguished by markedly higher expression of the uncharacterised gene Smp_161510, which was expressed along the dorso-ventral axis of 2-day old schistosomula (Fig. 2b). In adult worms, Smp_161510 did not exhibit dorso-ventral expression but was instead expressed throughout the worm body (Supplementary Fig. 4a) and co-localised with pan-muscle marker *troponin* (Smp_018250) (Fig. 2c and Supplementary Fig. 4b). A subset of cells in this muscle cluster also expressed *wnt-2* (Smp_167140) (Fig. 2a and Supplementary Fig. 3a). These *wnt-2* + cells showed an anterior-posterior gradient in two-day schistosomula (Fig. 2d) that remained consistent during the development from juveniles to mature adult worms (Fig. 2e, f and Supplementary Fig. 4a). Given that these markers showed distinct spatial distributions[50,55,56], we termed this population 'positional muscle'.

In a second muscle-like cluster (788 cells), an orthologue (Smp_167400) of the myoD transcription factor from *S. mediterranea* (dd_Smed_v6_12634_0_1)[32] was uniquely expressed (Fig. 2a). In addition, expression of *rhodopsin GPCR* (Smp_153210) was enriched in this *myoD*+ cluster (Fig. 2a and Supplementary Fig. 3a). Both genes showed a scattered expression pattern throughout the schistosomula (*myoD*; Supplementary Fig. 4c and *rhodopsin GPCR*; Fig. 2g) and, in adults, *myoD* (Smp_167400) is also scattered throughout the body (Fig. 2h and Supplementary Fig. 4d).

Finally, a third cluster (224 cells) of putative muscle cells was distinguished from the other clusters by its high *actin-2* (Smp_307020, Smp_307010) expression and lower expression of *myoD*, Smp_161510 and *rhodopsin GPCR* (Fig. 2a and Supplementary Fig. 3a). ISH confirmed *actin-2* expression throughout the body of the schistosomula and adults (Fig. 2i-k, Supplementary Fig. 4a). Our single-cell transcriptomic data suggested that *actin-2* was enriched but not specific to this cluster. In line with the transcriptome evidence, *actin-2* was expressed within cells from the other two muscle clusters (Supplementary Fig. 4e–i).

### Schistosomula have two distinct populations of tegumental cells.

We identified two populations of tegumental cells (Tegument 1 and Tegument 2; Fig. 3a and Supplementary Fig. 3b). The first tegumental cluster (Tegument 1, 182 cells) expressed several known tegument genes, including four that distinguish it from Tegument 2 (99 cells) and encode: Fimbrin (Smp_037230), TAL10 (Tegument allergen-like protein 10, Smp_074460), Annexin B2 (Smp_077720) and Sm21.7 (Smp_086480) (Fig. 3a and Supplementary Data 3)[7,43,44,57].

Fluorescently conjugated dextran specifically labels tegumental cell bodies[7]. Within the Tegument 1 population, cells expressing *annexin B2* were dextran+ (Fig. 3b), confirming that *annexin B2*

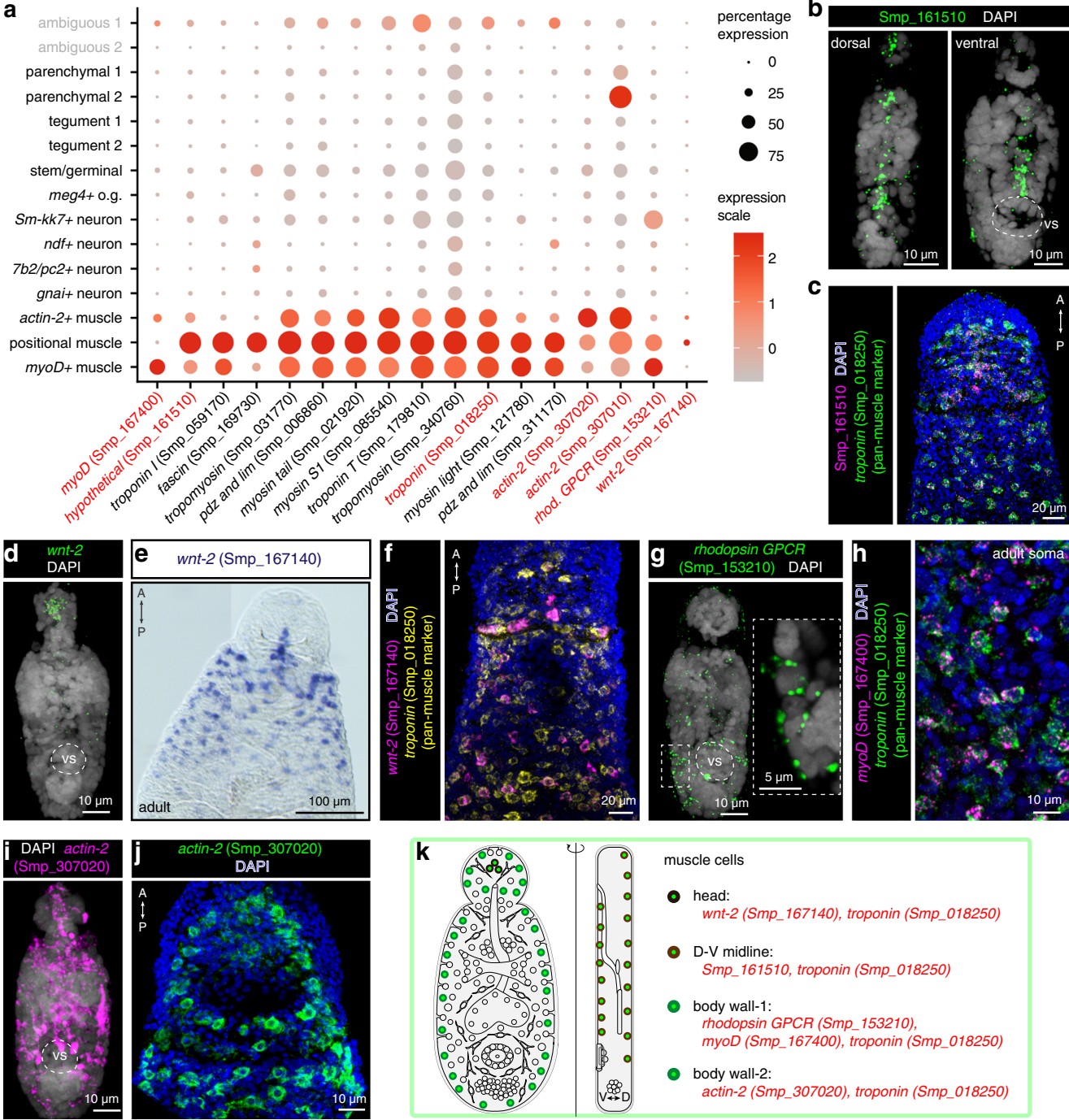

**Fig. 2 Muscle cells express positional information underlying parasite development. a** Expression profiles of cell markers that are specific or enriched in the muscle clusters. Genes shown in red were validated by ISH. **b** FISH of Smp_161510. Smp_161510-expressing cells are found in dorsal and ventral sides along the midline. vs: ventral sucker. Single confocal sections shown for each image. **c** Double FISH of Smp_161510 and a pan-muscle marker *troponin* (Smp_018250) in the head region of the adult worm. A: anterior; P: posterior. Maximum intensity projection (MIP) is shown. **d** FISH of *wnt-2* (Smp_167140) in 2-day old schistosomula, MIP. **e** Whole-mount in situ hybridisation (WISH) of *wnt* in the head region of the adult worm. The whole adult worm image is shown in Supplementary Fig. 4a. A: anterior; P: posterior. **f** Double FISH of *wnt-2* (Smp_167140) and *troponin* (Smp_018250) in the head region of the adult worm, MIP. **g** FISH of *rhodopsin GPCR* (Smp_153210). Left: MIP; right: single magnified confocal sections of the dotted box. **h** Double FISH of *myoD* (Smp_167400) and *troponin* (Smp_018250) in adult soma, MIP. **i, j** Spatial distribution of *actin-2* (Smp_307020) throughout the body of the parasite. **i** schistosomulum, MIP; **j** adult male, MIP. **k** Schematic that summarises the muscle cell types in 2-day old schistosomula. Marker genes identified in the current study are indicated in red. V: ventral; D: dorsal. The numbers of ISH experiments performed for each gene are listed in 'Methods' and Supplementary Data 7.

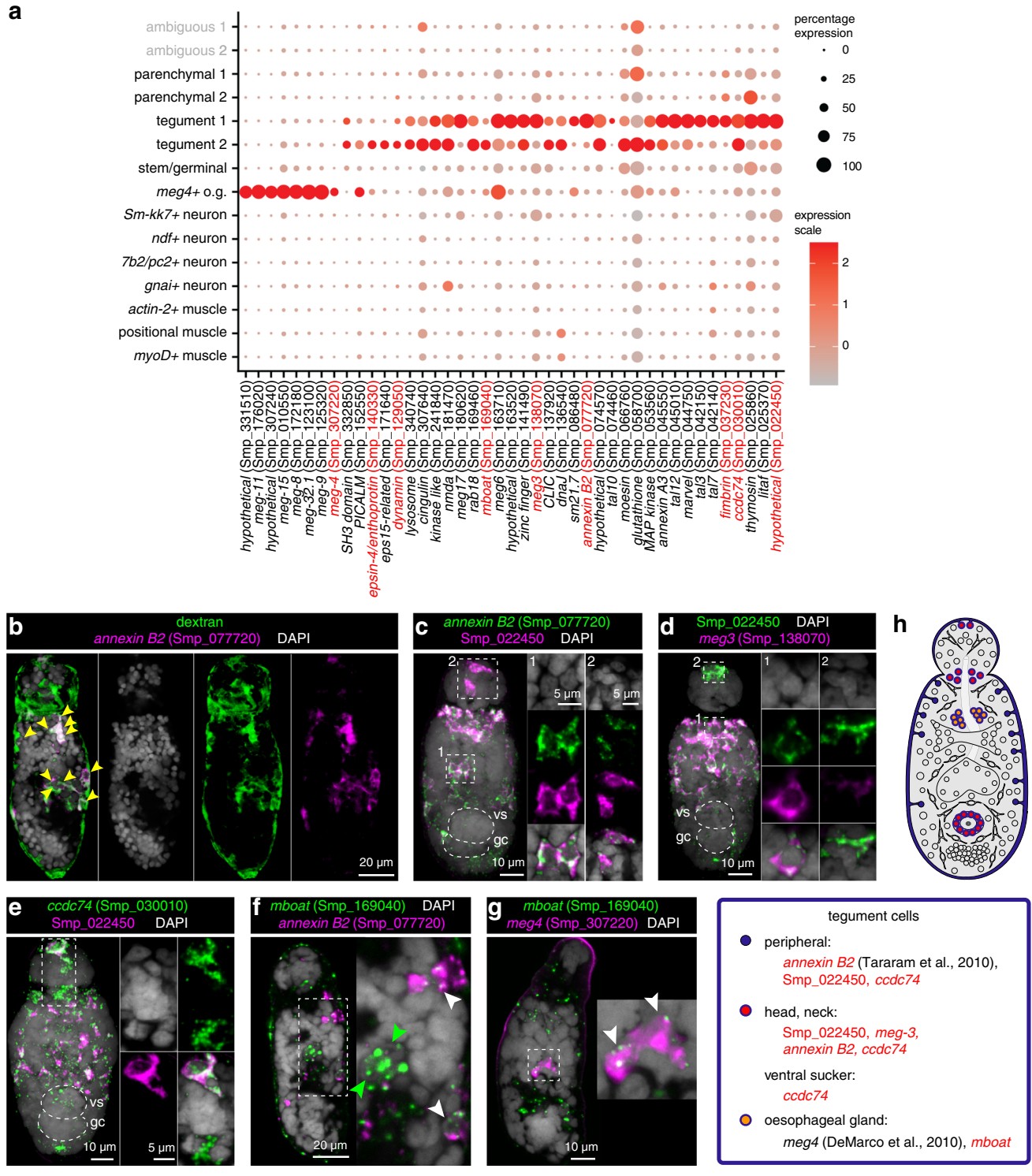

**Fig. 3 Two distinct populations of tegumental cells in schistosomula. a** Expression profiles of cell marker genes that are specific to or enriched in the tegument clusters. Genes validated by ISH are marked in red. **b** *annexin B2*+ cells have taken up the fluorescent dextran. Yellow arrowheads indicate double-positive cells. Single confocal sections are shown. **c** Double FISH of Tegument 1 markers *annexin B2* (Smp_077720) and Smp_022450. The majority of the cells show co-localisation (white signal). Left: MIP; right: zoomed in confocal sections. vs: ventral sucker; gc: germinal cell cluster. **d** Double FISH of Tegument 1 markers Smp_022450 and *meg3* (Smp_138070). The majority of the cells show co-localisation (white signal). **e** Double FISH of Tegument 1 marker (Smp_022450) with *ccdc74* (Smp_030010), MIP. The majority of cells show co-localisation (white signal), while a subset of cells in the neck region of the worm show single positive cells for Tegument 2 markers. **f**, **g** Double FISH of Tegument 2 marker *mboat* (Smp_169040) and **f** Tegument 1 marker *annexin B2* (Smp_077720) and **g** oesophageal gland marker *meg4* (Smp_307220), single confocal sections. Green arrowhead: *mboat*+ cells; white arrowhead: double-positive cells. **h** Schematic that summarises the tegument cell populations in 2-day old schistosomula. Marker genes identified in the current study are indicated in red. All previously reported genes are shown in black. The numbers of ISH experiments performed for each gene are listed in 'Methods' and Supplementary Data 7.

is a tegumental marker. The Tegument 1 population also showed enrichment for an uncharacterised gene (Smp_022450) that, to our knowledge, has not previously been reported as tegument-associated. Cells expressing Smp_022450 in the head, neck and body of the schistosomulum co-localised with *annexin B2* (Smp_077720) (Fig. 3c) and were dextran+ (Supplementary Fig. 5a). Tegument 1 also showed enrichment for the microexon gene *meg-3* (Smp_138070), with *meg3* co-localising with the novel tegument gene Smp_022450 in the neck and anterior region of the larva (Fig. 3d)

Distinguishing the second tegumental cluster was challenging due to a paucity of Tegument 2-specific markers (Fig. 3a and Supplementary Fig. 3b). Nonetheless, we selected two genes for further investigation: *ccdc74* (Smp_030010) and *mboat* (Smp_169040). We confirmed the tegumental assignment of *ccdc74* with dextran+ labelling[7] (Supplementary Fig. 5b) and a double FISH experiment showed colocalization of *ccdc74* with Tegument 1 marker Smp_022450 (Fig. 3e). However, distinct regions of expression for these two markers were also evident (Fig. 3e). Expression of *mboat* was lower but mostly distinct from Tegument 1 cells (Fig. 3f). Nonetheless, some level of co-localisation with Tegument 1 markers was still observed (Fig. 3f and Supplementary Fig. 5c). This is consistent with the expression profile for this gene that shows low-level expression in the Tegument 1 population (Fig. 3a and Supplementary Fig. 3b).

To explore more subtle differences in expression profiles between the two tegumental populations, we also investigated tentative functional differences. Analysis of marker genes enriched in Tegument 2 using the STRING database predicted a group of interacting genes involved in clathrin-mediated endocytosis[58] (Supplementary Fig. 5d, e). From this group, we chose *dynamin* (Smp_129050) and *epsin4* (Smp_140330) for FISH validation and found that they are expressed in regions of the schistosomula body, distinct from Tegument 1 cells (Supplementary Fig. 3b, 5f, g). In adult worms, we found that the spatial expression of many markers for Tegument 1 and 2 cells were similarly enriched in the anterior cell mass, ventral sucker, and throughout the worm body (Supplementary Fig. 6a, b).

**Micro-exon gene expression is enriched in the oesophageal gland**. We also discovered a small population of oesophageal gland cells (17 cells) that expressed *meg4* genes (Smp_307220/Smp_307240)[59] (Fig. 3a and Supplementary Figs. 3f and 6c, d). The oesophageal gland is an anterior accessory organ of the digestive tract[60] and is crucial for degradation of host immune cells and parasite survival[61]. This group of cells also expressed other *meg* genes with high specificity such as *meg8* (Smp_172180), *meg9* (Smp_125320), *meg11* (Smp_176020), *meg15* (Smp_010550) and *meg32.1* (Smp_132100) (Fig. 3a). The function of this class of genes is enigmatic but they have the capacity to generate protein diversity based on their propensity for exon skipping[59,62]. Given the expression of some *meg* genes around the oesophagus of adult parasites[63,64] and the developmental relationship between the oesophagus and the tegument[9,65], we tested if tegumental genes co-localised with any known genes from the *meg4+* oesophageal gland population (Fig. 3a). We found that *mboat* co-localised with the oesophageal gland marker *meg4* (Fig. 3g). Similarly to *mboat*, we also observed co-localisation of *epsin4* with *meg4* in the oesophageal gland (Supplementary Fig. 6d). In adults, Tegument 2 markers such as *epsin4* (Smp_140330) and *mboat* (Smp_169040) were consistently enriched in the oesophageal gland of the adult worm (Supplementary Fig. 6e, f). In the case of *mboat*, we could observe co-localisation with *meg4* in the oesophageal gland (Supplementary Fig. 6g). Therefore, the *meg4+* oesophageal gland cells share

similar molecular composition and function to the Tegument 2 cells (Fig. 3h).

**Identification of schistosome parenchymal and primordial gut cells**. Schistosomes, like other platyhelminthes, are acoelomates and lack a fluid-filled body cavity. Instead, their tissues are bound together by cells and extracellular matrix of the parenchyma[21,58]. We identified two cell types that most likely represent parenchymal cells (101 cells, Parenchymal 1; 57 cells, Parenchymal 2) that showed enriched expression of numerous enzymes such as lysosome, peptidase, and cathepsin (Fig. 4a and Supplementary Fig. 3c).

Cells expressing *cathepsin B* (Smp_141610) were spread throughout the worm parenchyma and showed long cytoplasmic processes stretching from each cell (Figs. 4b–f and Supplementary Fig. 7a, b). A similar expression profile was observed for *serpin* (Smp_090080) expressing cells in the later stages of schistosomula as well as in adult parasites (Supplementary Fig. 7c–e). In addition, parenchymal cells did not co-express markers that characterise other cell types, except for *actin-2* (*actin-2* muscle), which showed slight overlap in expression (Supplementary Fig. 7f–j).

In Parenchymal 2 cells, we found that *leucine aminopeptidase* (*lap*) (Smp_030000) was expressed in the primordial gut expressing *cathepsin B*' (Smp_103610)[66] and surrounding parenchymal tissue (Fig. 4g). Such mixed gut/parenchymal expression was also observed in adult parasites (Fig. 4h). This is consistent with previous studies in adult parasites where LAP was detected in the gut and in cells surrounding the gut[67]. Overall, the identified genes mark schistosomula parenchyma, while a few of them are also expressed in the gut primordia (Fig. 4i).

**Stem cells in two-day-old schistosomula**. Recently, it was shown that schistosomula carry two types of stem cell populations: somatic stem cells and germinal cells[16]. The somatic stem cells are involved in somatic tissue differentiation and homeostasis during intra-mammalian development, whereas the germinal cells are presumed to give rise to germ cells (sperm and oocytes) in adult parasites[16]. Less than 24 h after the cercaria enters the mammalian host to become schistosomulum, ~5 somatic stem cells at distinct locations begin to proliferate[16] (Fig. 5a). Germinal cells, on the other hand, are found in a distinct anatomical location called the germinal cell cluster, and only begin to proliferate ~1 week after penetrating the host[16].

We identified a single stem/germinal cell cluster (126 cells) that expressed the canonical cell cycle markers *histone h2a* (Smp_086860)[16] and *histone h2b* (Smp_108390)[7] (Fig. 5b and Supplementary Fig. 3d). In addition, this cluster also had a significant enrichment of translational components (Supplementary Fig. 2). We confirmed that *histone h2a* (Smp_086860) is expressed in ~5 cells, 1 medial and 2 on each side (Fig. 5a) and also in the germinal cell cluster a few days later (Supplementary Fig. 8a). In adults, *histone h2a* (Smp_086860) is expressed in somatic cells as well as in cells of the gonads (testis, ovary, and vitellaria) (Supplementary Fig. 8b). In addition, we identified a novel stem/germ cell marker *calmodulin (cam)* (Smp_032950). This gene was expressed similarly to *h2a*, but in some schistosomula, a few more *cam+* cells could be observed medially as well as near the germinal cell cluster (Fig. 5c). In addition, some *cam+* cells were also positive for *h2b* in schistosomula (Fig. 5d). In adults, *cam+* cells were expressed in the adult gonads (Fig. 5e) and soma (Fig. 5f).

In addition to *histone h2a* (Smp_086860), *histone h2b* (Smp_108390) and *cam* (Smp_032950), cells in this cluster expressed stem cell markers including *fgfrA* (Smp_175590) and

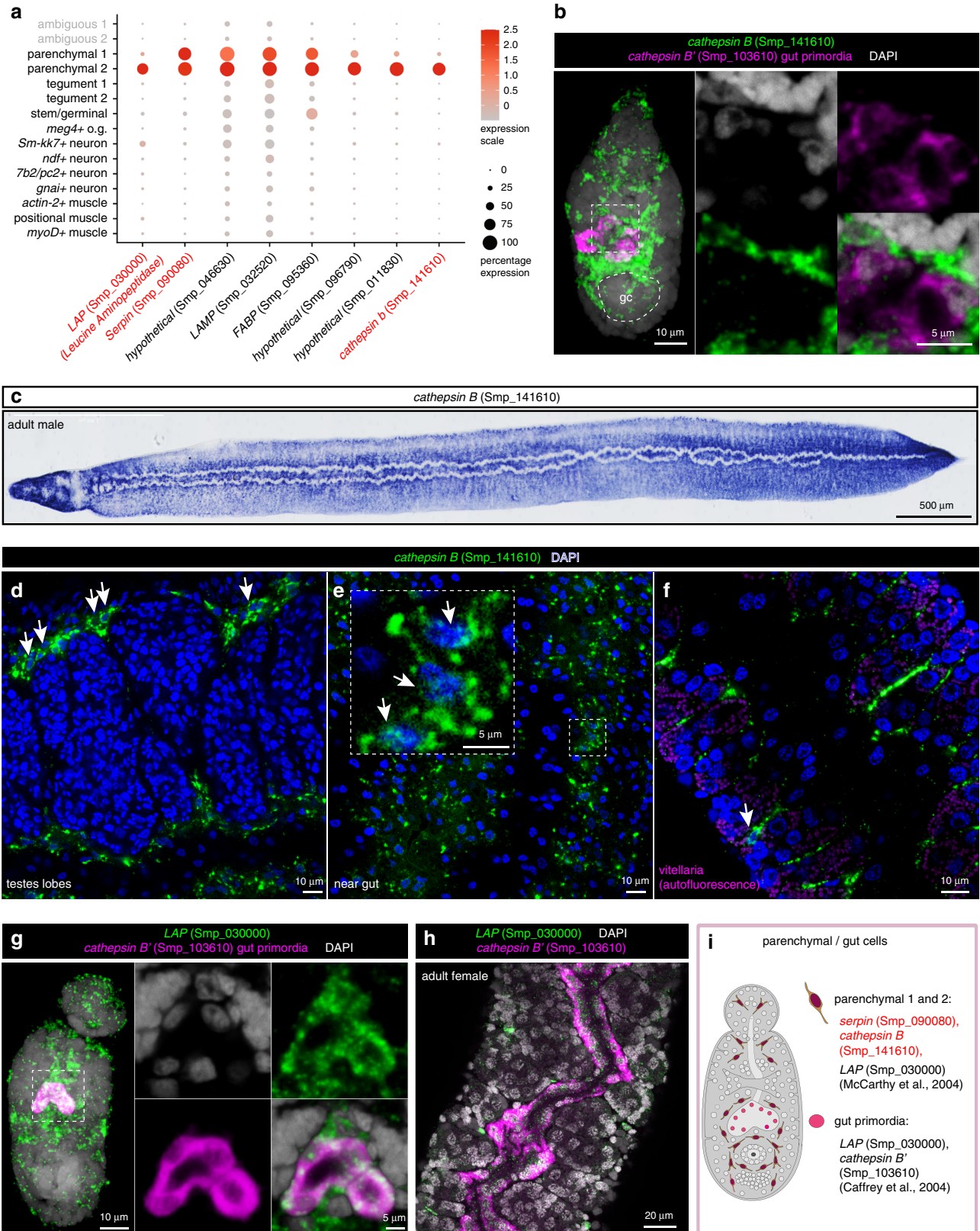

nanos-2 (Smp_051920)[15,16,68](Fig. 5b). Given that many of these genes have been associated with two distinct stem cell populations[16] (somatic and germinal), we tested if these cells could be further subclustered, but were unable to do so,

presumably due to the low expression level of some of these genes in most cells in this cluster (Supplementary Fig. 8c). Overall, these data suggest that this cluster does indeed represent population(s) of stem cells that might give rise to

**Fig. 4 Identification of schistosome parenchymal and primordial gut cells. a** Expression profiles of cell marker genes that are specific or enriched in the parenchymal clusters. Genes validated by ISH are marked in red. **b** Double FISH of parenchymal *cathepsin B* (Smp_141610) with a known marker of differentiated gut, *cathepsin B'* (Smp_103610), MIP. No expression of parenchymal *cathepsin B* is observed in the primordial gut. gc: germinal cell cluster. **c** WISH of parenchymal *cathepsin B* in adult males. **d**–**f** FISH of parenchymal *cathepsin B* in different regions of adult worms: **d** testes lobes, **e** gut, and **f** vitellaria. White arrows indicate positive cells. Single confocal sections shown. **g**, **h** *lap* (Smp_030000) is expressed in both parenchyma and in the **g** gut primordia as well as **h** adult gut, shown by double FISH with the gut *cathepsin B* (Smp_103610). **i** Schematic that summarises the parenchymal cell populations in 2-day schistosomula. Marker genes identified in the current study are indicated in red. All previously reported genes are shown in black. The numbers of ISH experiments performed for each gene are listed in 'Methods' and Supplementary Data 7.

somatic and germ cells during the course of parasite development within the mammalian host (Fig. 5g).

**Heterogeneity in cells of the schistosomulum nervous system.** Platyhelminthes have a central nervous system composed of cephalic ganglia and main nerve cords, and a peripheral nervous system with minor nerve cords and plexuses[11]. This system also plays a neuroendocrine role by releasing neuromodulators during development and growth[11,69–71].

We identified four distinct populations that expressed neural-associated genes (Fig. 6a and Supplementary Fig. 3e). One population (450 cells) was characterised by the expression of genes encoding neuroendocrine protein 7B2 (*7b2*, Smp_073270) and neuroendocrine convertase 2 (*pc2*, Smp_077980) and lack of *gnai* (Smp_246100) expression (Fig. 6a). FISH of *7b2* (Smp_073270) showed expression in cells of the cephalic ganglia in schistosomula. The cephalic ganglia region was identified using lectin succinylated Wheat Germ Agglutinin (sWGA)[12] staining. In adult worms, *7b2* was expressed in the cephalic ganglia as well as in the main and minor nerve cords (Fig. 6c–e). We refer to this cluster as '*7b2/pc2+* nerve' cells.

The second population (20 cells) expressed the uncharacterised gene Smp_203580 (Fig. 6f). Co-localisation experiments with *7b2* confirmed that this population was distinct from the central ganglia population (Fig. 6f). In the larvae, only six cells (two cells in the head and four cells in the body) expressed the novel marker Smp_203580 (Fig. 6f) but in adults, an expanded number of cells were found throughout the body of the parasite (Supplementary Fig. 9a–c). These cells displayed 2–3 long cellular processes, branching into different directions (Supplementary Fig. 9b). Interestingly, cells in this cluster also expressed the marker gene encoding KK7 (Smp_194830), known to be associated with the peripheral nervous system in *S. mansoni*[53] (Fig. 6g and Supplementary Fig. 9a, d). Therefore, we refer to this population as '*Sm-kk7+* nerve cells'.

The third population (141 cells) of cells expressed *gnai* (Smp_246100), a gene encoding a G-protein alpha subunit, group-I. FISH experiments showed expression of this gene in three cells: one in the gland region of the head, one in the neck region, and one in the body region (Fig. 6h). In adults, this gene is expressed around the main and minor nerve cords (Fig. 6i and Supplementary Fig. 9e, f). Some *gnai+* cells are also *7b2+* (Fig. 6i). We designated this population as '*gnai* + neurons'.

The last population comprised 32 cells and was annotated as *ndf+* neurons. This population was characterised by the expression of a *neurogenic differentiation factor* (*ndf*; Smp_072470) and a *neuropeptide receptor* (Smp_118040) (Fig. 6a). The *neurogenic differentiation factor* (*ndf*) Smp_072470 has recently been identified as a neuronal marker in adult schistosomes[72]. The orthologue of *ndf* in *S. mediterranea* (*neuroD1*) is also associated with neuronal populations[73]. Knockdown of this gene in combination with other neural specification genes in *S. mediterranea*, results in a decrease of 40% in a *npp-4+* population[73]. In addition, as well as expression in the nervous system, this gene is expressed in X1 neoblasts (stem

cells) from wounded *S. mediterranea* and in cells near regenerating anterior blastemas[74]. Although further experiments will be needed to ascertain the biological function of this population, data from adult schistosomes and *S. mediterranea* suggest this is indeed a neuronal population. Based on previous findings on *S. mediterranea*, it may be involved in the specification of other neural populations.

Overall, we show that in schistosomula, neuronal cells are transcriptionally and spatially heterogeneous (Fig. 6j), consistent with the pattern seen in more complex adult worms, where ~27 distinct neuron clusters could be identified in several anatomic regions[72].

**Conserved gene expression patterns in stem cells and neurons between *S. mansoni* and *Schmidtea mediterranea*.** Given that some of the populations described herein had not been previously characterised, we asked if we could further annotate our dataset by comparison to previously annotated single-cell RNAseq data from *Schmidtea mediterranea*, the closest free-living model organism to *S. mansoni*[32]. To compare clusters, we used a random forest (RF) model trained on *S. mediterranea* (Supplementary Fig. 10) to map gene expression signatures between both datasets[75]. Using the RF model, we classified each of the larval *S. mansoni* cells using the adult *S. mediterranea* labels. We discovered that the stem cell population in our dataset mapped to *S. mediterranea* neoblasts and progenitor clusters (Fig. 7). This is consistent with previous work that showed comparability between *S. mediterranea* and *S. mansoni* stem cells[6,15,16,47,68,76]. We found the strongest similarity between *Sm-kk7+* cells in schistosomula and the neuronal population annotated as *otoferlin 1* (*otf1+*) cells described by Plass et al.[32]. We also found other weaker signatures. For example, *NDF+* cells in *S. mansoni* mapped to *spp11+* neurons whilst tegument clusters mapped to epidermal neoblasts/progenitors in *S. mediterranea*. In addition, we observed that some cells on the schistosomula muscle populations mapped to *S. mediterranea* muscle progenitors. Taken together, these results suggest that despite great differences in developmental stages between larval schistosomula and the asexual adult *Schmidtea mediterranea* used for this comparison, marker genes for stem cells and neuronal populations have been conserved (Fig. 7 and Supplementary Data 5 and 6).

## Discussion

In this study, we have generated a cell atlas of the schistosomulum, the first intra-mammalian developmental stage of the blood fluke *S. mansoni* and a key target for drug and vaccine development[77,78]. A goal of single cell sequencing is to capture the heterogeneity of all cells and classify them into their broad types. Stochastic gene expression fluctuations mean that cells of the same cluster do not necessarily display the same expression profiles. Our transcriptome analysis enabled the conservative characterisation of 13 distinct clusters, with sufficient sensitivity to detect as few as three cells per parasite, as demonstrated by the

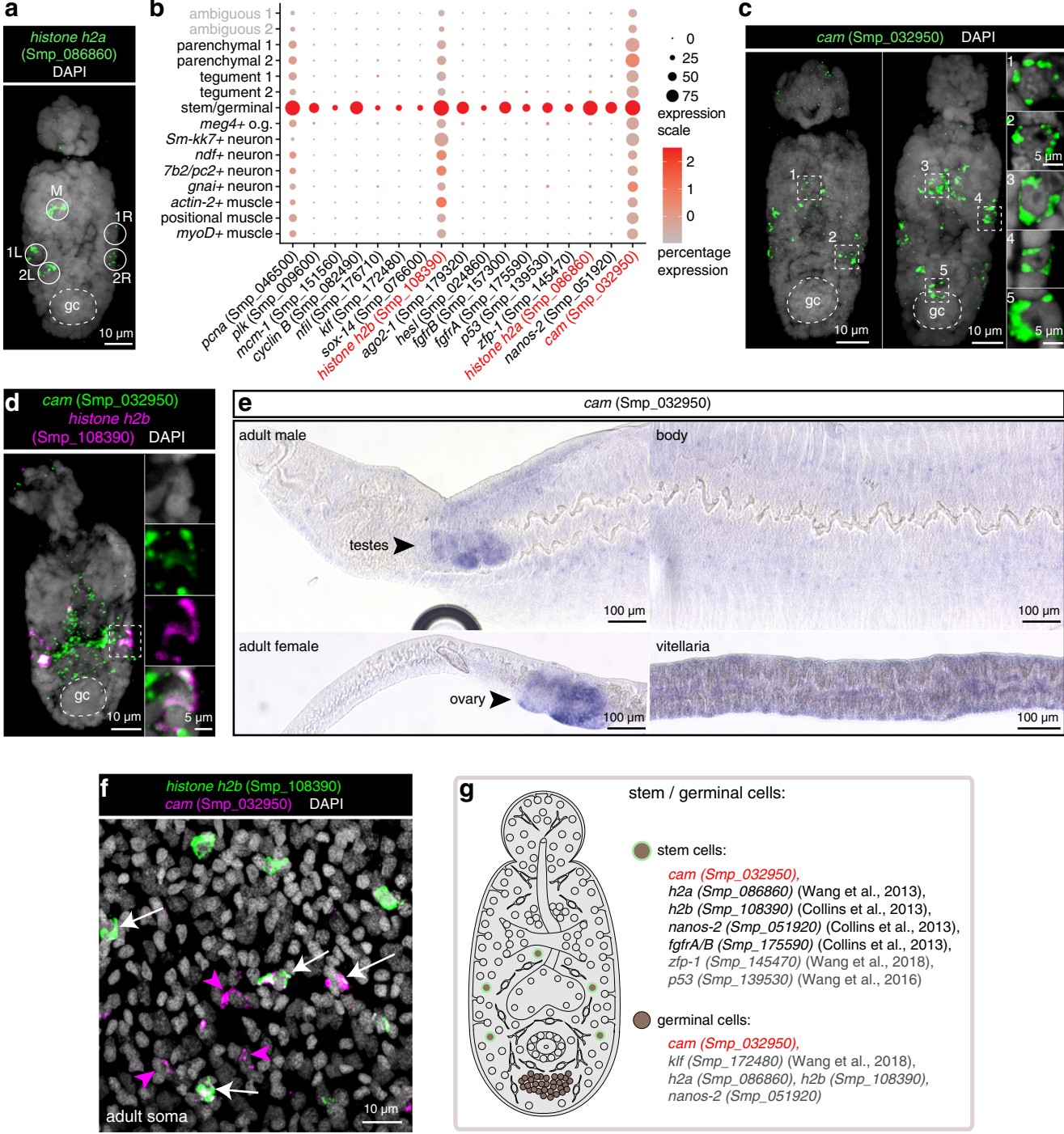

**Fig. 5 A single cluster of stem cells in 2-day old schistosomula. a** FISH of *h2a* (Smp_086860) shows ~5 stem cells located at distinct locations—1 medial cell (M) and 2 lateral cells on each side (1L and 2L, 1R, and 2R; L: left; R: right), MIP. **b** Expression profiles of cell marker genes that are specific or enriched in the stem/germinal cell cluster. Genes validated by ISH are marked in red. **c** FISH of *cam* (Smp_032950) shows a similar localisation pattern as *h2a*, with some worms with a few more *cam*+ cells in the medial region as well as in the germinal cell cluster region, MIP. gc: germinal cluster. **d** Double FISH of *cam* (Smp_032950) and a previously validated schistosome stem cell marker *h2b* (Smp_108390), MIP. **e** WISH of *cam* (Smp_032950) in adult parasites shows enriched expression in the gonads including testis, ovary, and vitellarium, as well as in the mid-animal body region. **f** Double FISH of *cam* and *h2b* in adult soma. A single confocal section is shown. White arrows indicate co-localisation of two genes and magenta arrowheads indicate cells expressing only *cam*. **g** Schematic that summarises the stem and germinal cell populations in 2-day old schistosomula. Marker genes identified in the current study are indicated in red. All previously reported genes are shown in black. Genes that are enriched in this cluster but have not been directly shown by ISH are shown in grey. The numbers of ISH experiments performed for each gene are listed in 'Methods' and Supplementary Data 7.

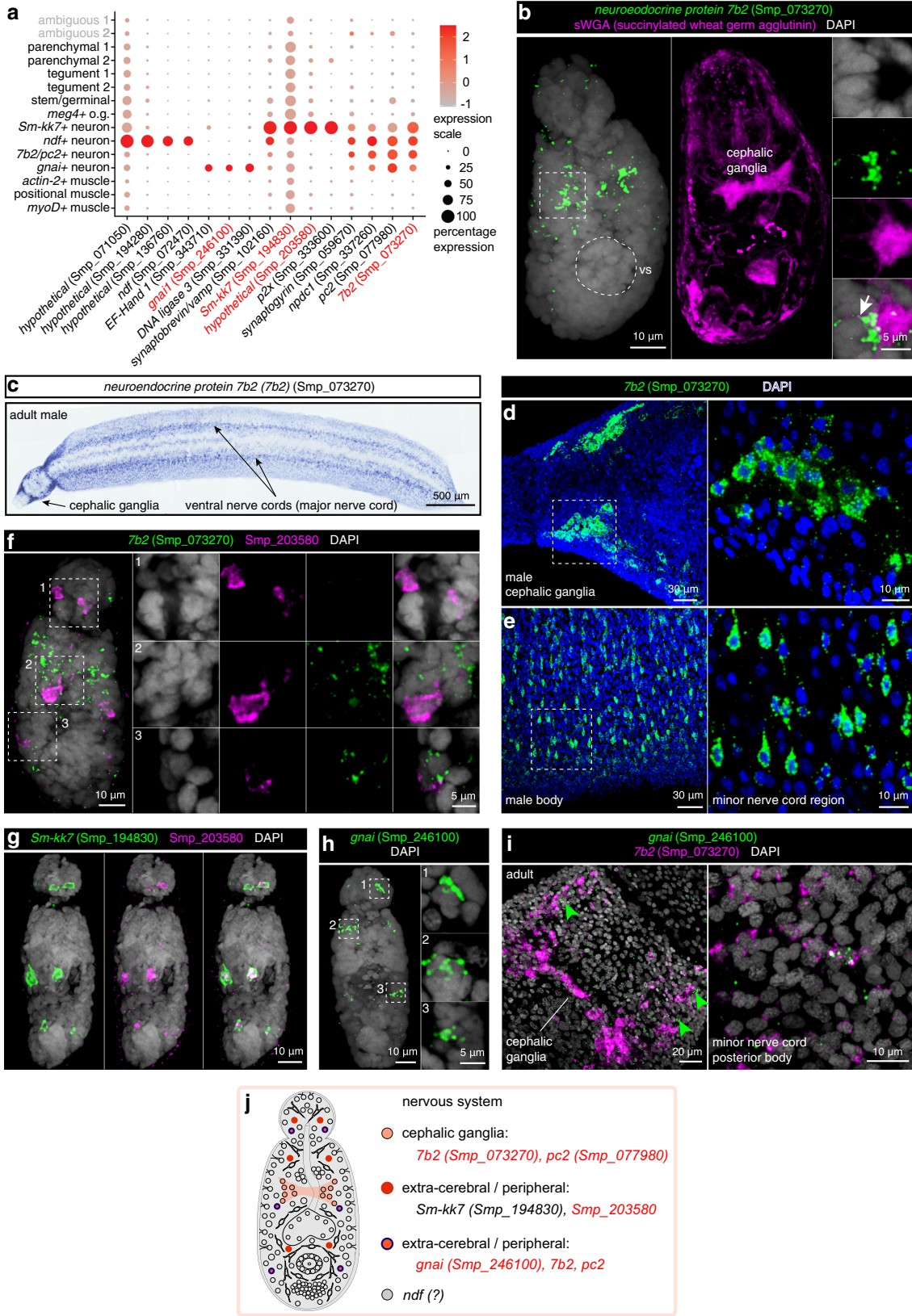

ISH experiments. Importantly, the latter allowed us to validate key marker genes for each of the cell clusters, spatially mapping the cell populations in both schistosomula and adult worms and linking transcriptomic profiles to anatomical features of the organism.

By determining the transcriptome of individual cells from schistosomula, we uncovered marker genes not only for known populations, such as stem and tegument cells, but also for previously undescribed cell clusters, such as parenchymal cells. We found that marker genes of the parenchymal tissue are also

**Fig. 6 Heterogeneity in cells of schistosomula nervous system. a** Expression profiles of cell marker genes that are specific or enriched in the neuronal clusters. Genes validated by ISH are marked in red. **b** Cephalic ganglia marked by sWGA lectin shows co-localisation with *7b2* (Smp_073270)(white arrow), MIP. **c** WISH of *7b2* (Smp_073270) in adult male. **d, e** FISH of *7b2* (Smp_073270) in **d** cephalic ganglia and **e** body region of adult worms, single confocal sections. **f** Double FISH of *7b2* (Smp_073270) and Smp_203580 shows that six cells that are Smp_203580+ do not co-localise with *7b2*+ cells. **g** Double FISH of Smp_203580 with *Sm-kk7* (Smp_194830). All Smp_203580+ cells co-localise with *Sm-kk7*, MIP. **h** *gnai* (Smp_246100) FISH shows expression in a few cells along the anterior-posterior axis of the somule, MIP. **i** Double FISH of *gnai* and *7b2* shows some co-localisation in the nerve tracts in an adult male. Single confocal sections are shown. **j** Schematic that summarises the neuronal cell populations in two-day schistosomula. Marker genes identified in the current study are indicated in red. All previously reported genes are shown in black. The numbers of ISH experiments performed for each gene are listed in Methods and Supplementary Data 7.

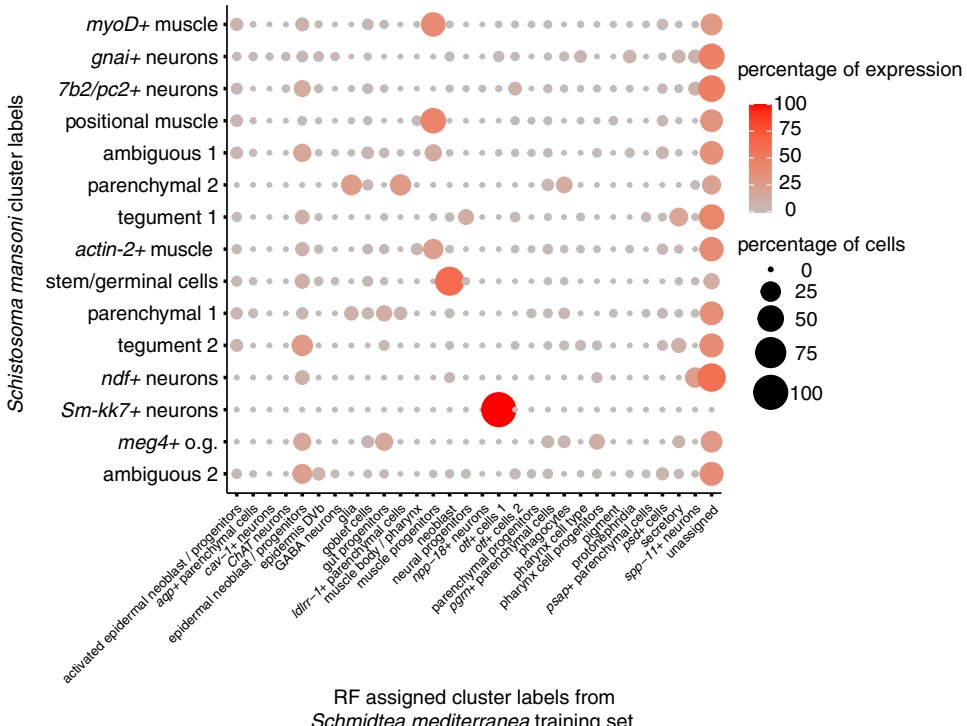

**Fig. 7 Gene-expression patterns of stem cells and neurons conserved between *S. mansoni* and *Schmidtea mediterranea*.** Random forest classifier used to assign cells from schistosomulum clusters into categories based on a *Schmidtea mediterranea* scRNAseq dataset[32]. The colours and size of the circles represent the proportion of cells from each cluster (y-axis) that matches each *S. mediterranea* category label (x-axis). Only categories that received a maximum vote by a margin of >16% of trees during the prediction are included. Cells that did not fit any classification were classed as 'not assigned'.

expressed in the primordial gut. However, the relationship between the parenchyma and gut primordial cells is yet to be determined. In planarians, the orthologous *cathepsin* gene (dd_Smed_v6_81_0_1) is a marker for cathepsin+ cells that include cells in the parenchyma[32,33]. This planarian *cathepsin* (dd_Smed_v6_81_0_1) is also expressed in the intestine[33] and gut phagocytes[32,33]. Similarly, planarian aminopeptidase (dd_Smed_v6_181_0_1) is expressed in *cathepsin*+ cells, epithelia and intestine[32,33]. Thus, further work is required to characterise schistosome parenchymal cells and their signaling mechanisms with the surrounding gut cells[79].

Previously, *S. mansoni* cell types have been revealed primarily through a combination of morphological and ISH studies of specific tissues, with stem and tegument cell populations being among the best characterised[6,7,15,16]. In the present study, we have identified and validated new markers, including a novel stem cell marker *calmodulin* (Smp_032950) that, to our knowledge, has not previously been associated with stem cells. Calmodulins are $Ca^{2+}$ binding proteins involved in the miracidium-to-sporocyst transition, sporocyst growth[80] and egg hatching[81]. In addition, we found this calmodulin-encoding gene to be expressed in the reproductive organs of adult males and females. In contrast, we

were unable to identify three stem cell populations (*delta*, *kappa* and *phi*) that were previously described by Wang et al.[16]. In the latter study, marker genes were identified from the single-cell transcriptomes of 35 cells obtained from sporocyst germinal centres. Some marker gene expression was subsequently confirmed in 2-day old schistosomula. In our study, a particular cell population was not specifically targeted. As such, our sensitivity to identify the reported germline subcluster markers may have been reduced, particularly given their expected low expression levels[16].

Coordinated neuromuscular activity is essential for schistosomes to migrate through host tissue[82]. Although circular and longitudinal muscle layers have been described in *S. mansoni*[10,12,82], we found no evidence that the three muscle clusters correspond to different anatomical fibre arrangements. In the free-living planarian *S. mediterranea*, a population of muscle cells also shows no specific muscle layer localisation, but instead forms a cluster based on enriched expression of position-control genes (PCGs)[33,83]. We therefore reasoned that this may be the case for at least some of the muscle cells in our dataset.

Previous studies have shown that numerous vesicles are produced by endocytosis from cell bodies and trafficked to the

syncytial cytoplasm of the tegument[84,85]. Our analysis revealed two distinct populations of tegumental cells, with a potential involvement of one of these populations (Tegument 2) in producing vesicles. By analysing inferred interactions between Tegument 2 genes, a group was identified that included homologues of known vesicular transport proteins: epsins and a phosphatidylinositol-binding clathrin assembly protein. Further, the most discriminatory marker that we found for the Tegument 2 cluster likely acylates membrane lysophospholipids[86,87]. It is tempting to speculate that mboat acylates the specific phosphatidylinositol membrane phospholipids required for clathrin-related endocytosis. We also show through the FISH validation some level of co-localisation of tegumental genes with meg4 + oesophageal gland. Oesophagus connects the mouth to the gut and is surrounded by a gland that secretes, amongst other things, proteins encoded by meg genes that help process the ingested blood[63,88] by degrading immune cells and preventing them from entering the gut[61]. They are therefore crucial for parasitic development and a prime target for vaccine development[63].

Knowledge of planarian stem cells has previously informed the study of stem cells in S. mansoni[68]. Our comparison between schistosomula and S. mediterranea clusters uncovered conserved features for stem cells and neurons and served to support cell type assignment in schistosomula. Given that nerve cell populations have remained poorly characterised at the transcriptome level in schistosomes, planarians may serve as a model to understand the nervous system biology in schistosomula. A particularly remarkable feature of planarian biology is their regenerative properties. An individual worm comprises all cell types at intermediate stages of development and regeneration[32,33,89]. This has enabled recent single-cell sequencing studies in planarians to characterise developmental trajectories from within the soma of adult worms[32]. However, schistosomes do not share this regenerative property with their free-living relatives, instead intermediate stages of schistosome development necessarily need to be captured. The data from the present study represent the first logical step in that characterisation.

In characterising previously unknown marker genes and cell types, we have been careful to validate our findings against known markers where possible. Signals from damaged or dying cells caused by laboratory procedures are challenging to eliminate[90,91] but we have followed a stringent FACS-based selection protocol to enrich for live cells. Like others, we have looked at the expression of mitochondrial genes and stress-related genes, as recommended for single-cell sequencing analysis[92] and our bioinformatic findings are supported by FISH-based validation in both schistosomula and adult worms. Notwithstanding these measures, some known cells were not detected, possibly due to their rarity in the schistosomula or fragility during tissue dissociation. The absence of a distinct cluster of protonephridia cells, known to be present in schistosomula[12,93], is a notable example. In our data, the S. mansoni bone morphogenic protein (BMP) homologue (Smp_343950)[94], a previously described protonephridial marker, was found in the muscle and nervous system (Supplementary Fig. 11). Previous single-cell studies in S. mediterranea have found that relatively rare cell types are sometimes embedded in larger neuronal clusters[32,33], and therefore, it is possible that this is also the case for this cell group.

We necessarily focussed on gene expression changes amongst protein coding genes because these are now well annotated on the reference genome and therefore can be accurately quantified. This is an essential first step in unravelling the developmental biology of this important parasite. Long non-coding RNAs (lncRNA) in S. mansoni have also been identified from transcriptomic datasets[95]. As the definitions of these additional RNA genes are included into the reference genome annotation, future reanalyses of our

single-cell data may add further dimensions to the transcriptional dynamics and identify further markers of developing cells, tissues or indeed new cell types. Overall, our study demonstrates the power of single-cell sequencing, coupled with ISH validation, to transcriptionally and spatially characterise cell types of an entire metazoan parasite for the first time.

## Methods

**Ethics statement.** The complete life cycle of Schistosoma mansoni (NMRI strain) is maintained at the Wellcome Sanger Institute (WSI). Balb/C female mice, 8–12 weeks old by the time of infection, are used as definitive hosts. The mouse infections at the WSI were conducted under Home Office Project Licence No. P77E8A062 held by GR, and all protocols were presented and approved by the Animal Welfare and Ethical Review Body (AWERB) of the WSI. The AWERB is constituted as required by the UK Animals (Scientific Procedures) Act 1986 Amendment Regulations 2012. To harvest parasites for validation using in situ hybridization, we used Swiss-Webster (Taconic Biosciences) female mice that are between 5 to 12 weeks of age at the time of infection. The mouse infection was done using S. mansoni (NMRI strain received from Biomedical Research Institute (Rockville, MD)) and all mice were handled in accordance with the Institutional Animal Care Use Committee protocol at the University of Wisconsin-Madison (M005569).

**Preparation of parasites.** S. mansoni schistosomula were obtained by mechanical transformation of cercariae and cultured[96]. In brief, experimentally-infected snails were washed, transferred to a beaker with water (~50-100 ml) and exposed under light to induce cercarial shedding for two hours, replacing the water and collecting cercariae every 30 min. Cercarial water collected from the beaker was filtered through a 47 μm stainless steel Millipore screen apparatus into sterile 50 ml-Falcon tubes to remove any debris and snail faeces. The cercariae were concentrated by centrifugation ($800 \times g$ for 15 min), washed three times in 1× PBS supplemented with 2% PSF (200 U/ml penicillin, 200 μg/ml streptomycin, 500 ng/ml amphotericin B), and three times in 'schistosomula wash medium' (DMEM supplemented with 2% PSF and 10 mM HEPES (4-(2-hydroxyethyl)-1-piperazineethanesulfonic acid)). The cercarial tails were sheared off by ~20 passes back and forth through a 22-G emulsifying needle, schistosomula bodies were separated from the sheared tails by Percoll gradient centrifugation, washed three times in schistosomula wash medium and cultured at 37 °C in modified Basch's medium under 5% $CO_2$ in air[96].

**Single-cell tissue dissociation.** Two days after transformation, the schistosomula cultured in modified Basch's media at 37 °C and 5% $CO_2$ were collected and processed in two separate batches (batch1 and batch2). Schistosomula collected from two different snail batches were considered biological replicates. Data collected as batch3 are 'technical' replicates of batch2 given they were collected on the same day and from the same pool of parasites. In each experiment, approximately 5000 larvae were pooled in 15 ml tubes and digested for 30 min in an Innova 4430 incubator with agitation at 300 rpm at 37 °C, using a digestion solution of 750 μg/ml Liberase DL (Roche 05466202001) in PBS supplemented with 20% FBS. The resulting suspension was passed through 70μm and 40μm cells strainers (Falcon). Dissociated cells were spun at 300 rpm for 5 mins and resuspended in 1× cold PBS supplemented with 20% heat inactivated fetal bovine serum (twice). The resulting cell suspension was co-stained with 0.5 μg/ml of Fluorescein Diacetate (FDA; Sigma F7378) to label live cells, and 1 μg/ml of Propidium Iodide (PI; Sigma P4864) to label dead/dying cells, and sorted into eppendorf tubes using the BD Influx™ cell sorter by enriching for FDA +/ PI− cells[97]. It took 2–3 h from the enzymatic digestion to generating single-cell suspensions ready for library preparation on the 10X Genomics Chromium platform.

**10X Genomics library preparation and sequencing.** The 10X Genomics protocol ("Single Cell 3′ Reagent Kits v2 User Guide" available from https://support.10xgenomics.com/single-cell-gene-expression/index/doc/user-guide-chromium-single-cell-3-reagent-kits-user-guide-v2-chemistry) was followed to create gel in emulsion beads (GEMs) containing single cells, hydrogel beads and reagents for reverse transcription, perform barcoded cDNA synthesis, and produce sequencing libraries from pooled cDNAs. The concentration of single cell suspensions was approximately 500 cells/μl, as estimated by flow cytometry-based counting, and cells were loaded according to the 10X protocol (Chromium Single Cell 3′ Reagent Kits v2), intended to capture approximately 7000 cells per reaction. However, after sequencing and preliminary analysis, we found the actual number of captured cells was closer to ~1200 cells per experiment. Library construction (following GEM breakage) was using 10X reagents following the "Single Cell 3′ Reagent Kits v2 User Guide". The libraries were sequenced on an Illumina Hiseq4000 (paired-end reads 75 bp), using one sequencing lane per sample. All raw sequence data was deposited in the ENA under the project accession ERP116919.

**Protein-coding genes.** Schistosoma mansoni gene annotation is based on the version 7 (v7) genome assembly (https://parasite.wormbase.org/Schistosoma_mansoni_prjea36577). The identifiers for all genes contain the Smp_ prefix followed by a

unique 6-digit number; entirely new gene models have the first digit '3', eg. Smp_3xxxxx. To assign a gene name and functional annotation (used in Supplementary Data 2) to 'Smp_' identifiers, protein-coding transcript sequences were BLAST-searched against SwissProt3 to predict product information (blastp v2.7.0). Some genes also maintained previous functional annotation from GeneDB. Genes lacking predicted product information were named *hypothetical* genes.

**Mapping and quantification of single-cell RNA-seq.** Single-cell RNA-seq data were mapped to the *S. mansoni* reference genome v7 (https://parasite.wormbase.org/Schistosoma_mansoni_prjea36577) using the 10X Genomics analysis pipeline Cell Ranger (v 2.1.0). We relied on the default cut-off provided by Cell ranger to detect empty droplets. Approximately 67% of sequenced reads mapped confidently to the transcriptome with an average 297,403 reads per cell. In total 3513 cells were sequenced, with a median 918 genes expressed per cell.

**Clustering using Seurat.** The Seurat package (version 3.1.5) (https://satijalab.org/seurat/) was used to analyse the raw values of the matrix[41]. We removed cells that had greater than 30,000 Unique Molecular Identifiers (UMIs) and less than 600 genes per cell. We also removed cells with mitochondrial expression percentage (MT) > 2.5%. We normalised using the NormalizeData function from Seurat (http://satijalab.org/seurat/). Following normalisation, we identified 2000 highly variable genes using the Seurat FindVariableGenes function. We employed two methods to determine the number of PCs for clustering. The first, was the visual inspection of the ElbowPlot as provided by Seurat. The second method uses molecular cross-validation[98] to determine the optimal number of PCs required to cluster the dataset (https://github.com/constantAmateur/MCVR/blob/master/code.R). We identified 15 clusters (including two ambiguous clusters) using the FindClusters function from Seurat using the first 25 PCs.

**Identifying marker genes and cluster annotation.** To annotate each cluster, we manually inspected the top markers for each of the populations (Supplementary Data 2) and compared to the top markers curated from the literature (Supplementary Data 3). We used the Seurat package to identify marker genes for each population using the function FindAllMarkers and test.use = "roc", only.pos = TRUE, return.thresh = 0, as specified in the Seurat best practices (https://satijalab.org/seurat/). We used the 'area under the ROC' (AUC) > 0.7 value and spatial information of those genes to determine the identity of a specific population. We characterised 13 populations using gene annotations and spatial information.

**Gene ontology (GO) analysis.** The Gene Ontology (GO) annotation *for Schistosoma mansoni* was obtained by running InterProScan v5.25-64.0 (https://www.ebi.ac.uk/interpro/). GO term enrichment was performed using the weight01 method provided in topGO[99] v2.34.0 (available at http://bioconductor.org/packages/release/bioc/html/topGO.html) for all three categories (BP, MF, and CC). For each category, the analysis was restricted to terms with a node size of > =5. Fisher's exact test was applied to assess the significance of overrepresented terms compared with all expressed genes. The threshold was set as FDR < 0.01.

**STRINGdb analysis.** We used STRINGdb[100] to identify possible gene interactions that would enable us to differentiate between tegumental clusters. Briefly, the *S. mansoni* V7 gene identifiers for the tegument 2 cluster with AUC ≥ 0.7 in Seurat were converted to *S. mansoni* V5 gene identifiers. The V5 gene identifiers were analysed in STRINGdb v11.0[100]. Human, *Caenorhabditis elegans* and *Drosophila melanogaster* orthologues of these genes were identified from WormBase ParaSite[101].

**Finding Schmidtea-Schistosoma orthologues for random forest analysis.** We accessed the transcriptome reference (version 6) for the asexual strain of *Schmidtea mansoni* from planmine[102]. This version is a Trinity de novo transcript assembly[103]. We used orthoMCL[104] to find one-to-one orthologues between *S. mediterranea* and *S. mansoni* as follows: (i) Smp and dd_Smed gene identifiers were collapsed to their root names (a single spliceform was taken for each gene) and clusters chosen with a single *Schmidtea* and *Schistosoma* gene; (ii) *Schistosoma* genes present on haplotypic contigs were removed where applicable to reduce multiple gene sets to a single copy; and (iii) Single representative *Schmidtea* genes were randomly selected from orthologue groups containing many *Schmidtea* and only a single *Schistosoma* gene and where there was no mapping to another orthologue cluster. This gave us a set of *Schmidtea-Schistosoma* orthologous gene-pairs. All *Schistosoma* genes were then replaced in the *Schistosoma* single-cell matrix with their *S. mediterranea* orthologues.

**Preparing and annotating *S. mediterranea* single-cell data for use with random forest classifier.** A single-cell dataset published for *Schmidtea mediterranea* comprising 21,612 cells generated using a droplet-based platform[32] was employed for this analysis. The relevant files were downloaded from https://shiny.mdc-berlin.de/psca/. The Seurat package (version 3.1.5) was used for all analysis of the *Schmidtea* dataset (https://satijalab.org/seurat/). We only kept cells that expressed

at least 200 genes, in a minimum of 3 cells. After QC, 21,612 cells and 28,030 transcripts remained. We normalised using the NormalizeData function from the Seurat (http://satijalab.org/seurat/). Following normalisation, we identified highly variable genes using the Seurat FindVariableGenes function. We assigned identities to cells based on the categories from Plass, et al.[32] but with the following changes to yield a total of 30 categories:

a. Neoblasts 1–13 grouped together as a single category labelled as "neoblasts".
b. Activated early epidermal progenitors and epidermal neoblasts combined into "activated epidermal neoblast/progenitors".
c. Chat neurons 1–2 combined into "Chat neurons".
d. Early/Late epidermal progenitors and epidermal DVb neurons combined into "epidermal DVb neoblast/progenitors".
e. Secretory 1–4 combined into "secretory".

**Evaluating the random forest on the Schmidtea dataset and applying it to *S. mansoni*.** We first evaluated the random forest (RF) classifier on the *Schmidtea* dataset of 21,612 cells using a set of 692 genes that were identified as Variable Genes by the Seurat function (FindVariableGenes) and had one-to-one orthologues in the *Schmidtea* and *Schistosoma* datasets. We used the R package randomForest (version 4.6-14) to aggregate scores from 500 decision trees built from a subset of the data. The training set comprised cells belonging to each of the 30 *Schmidtea* populations annotated from Plass, et al.[32] as described in the section above, with a maximum of 70% of cells per cluster. The remaining 30% was used for testing how well the training set could assign labels. We assigned a class to each cell when a minimum of 16% of trees in the forest converged onto a decision. When no class could be assigned, the cells and therefore the clusters where they belong were labelled as 'not assigned'. Using the RF package[105], the RF decision trees from the *Schmidtea* training set, built on 825 *S. mediterranea* genes were then used to assign labels to the *S. mansoni* cells.

**In situ hybridization (ISH).** Fluorescence in situ hybridization (FISH) and whole-mount colorimetric in situ hybridization (WISH) were performed following previously established protocols[15,16,47] with modifications specific to schistosomula. Schistosomula were killed with ice-cold 1% HCl (VWR, JT9535-3) for 30–60 s before fixation. Schistosomula were fixed for ~0.5–1 hour at room temperature in 4% formaldehyde, 0.2% Triton X-100 (Fisher, BP151-500), 1% NP-40 (Fluka, 74385) in PBS. Adult parasites were fixed for 4 h in 4% formaldehyde in PBSTx (1× PBS + 0.3% Triton X-100) at room temperature. After fixation, schistosomula and adults were dehydrated in methanol and kept in −20 °C until usage. Parasites were rehydrated, permeabilised by 10 μg/mL proteinase K (ThermoFisher, 25530049) for 10–20 min for schistosomula or 20 μg/mL proteinase K for 30 min for adults, and fixed for 10 min immediately following proteinase K treatment.

For hybridization, labelled FISH and WISH riboprobes were generated using either DIG (digoxigenin)-12-UTP (Sigma, 11209256910), DNP (dinitrophenol)-11-UTP (PerkinElmer, NEL555001) or fluorescein-12-UTP (Sigma, 11427857910). DIG-riboprobes were used for single FISH and WISH, and FITC- and DNP- riboprobes were used for double FISH. Anti-DIG-POD (1:500–1:2000, MilliporeSigma, 11207733910), anti-FITC-POD (1:500–1:2000, MilliporeSigma, 11426346910), anti-DNP-HRP (1:500 of 0.25 mg/ml (0.5 μg/ml), custom made from Vector Laboratories) antibodies were used for FISH and anti-DIG-AP (MilliporeSigma, 11093274910) antibody was used at 1:2000 for WISH. Anti-DIG-POD and anti-DIG-AP antibodies were incubated in FISH blocking solution (5% horse serum (Sigma, H1270-500ML) and 0.5% Western Blocking Reagent (Roche, 11921673001) in TNTx (100 mM Tris pH 7.5, 150 mM NaCl, 0.3% Triton X-100)) overnight at 4 °C and anti-FITC-POD and anti-DNP-POD was incubated for a total of ~4 h at room temperature before or after overnight incubation at 4 °C. Tyramide conjugates were synthesized from N-hydroxy-succinimydyl esters of 5-(and-6)-carboxytetramethylrhodamine (TAMRA) (Molecular Probes) or DyLight 633 (Pierce). For tyramide signal amplification, fluorophore-conjugated tyramide (TAMRA or DyLight 633) diluted 1:250–1:500 in 100 mM borate buffer pH 8.5, 2 M NaCl, 0.003% $H_2O_2$, and 20 μg/ml 4-iodophenylboronic acid. For double FISH experiments, residual peroxidase activity was quenched by incubating for 45 min in 100 mM sodium azide (Fisher) diluted in PBSTx. Primers used for cloning a fragment of marker genes and riboprobe generation are listed in Supplementary Data 4.

For the gene *MyoD*, a probe, buffers and hairpins for third generation in situ hybridization chain reaction (HCR) experiments were purchased from Molecular Instruments (Los Angeles, California, USA). Schistosomules were fixed as described above and mounted using DAPI fluoromount-G (Southern Biotech). Experiments were performed following the protocol developed for whole-mount nematode larvae[106] and imaged on a confocal laser microscope (Sp8 Leica).

**Immunostaining and lectin labelling.** For lectin labeling, fluorescein succinylated wheat germ agglutinin (sWGA) (Vector Labs) was used at 1:500 dilution in FISH blocking solution overnight at 4 °C.

**Dextran labelling.** Fluorescent dextran was used to label tegument cells[7]. Briefly, schistosomula were transferred to 20 μm mesh in order to flush out as much media while retaining parasites inside the mesh. 2.5 mg/ml dextran biotin-TAMRA-

dextran (ThermoFisher Scientific, D3312) was added to the mesh and parasites transferred into a 1.7 ml tube. Immediately after the transfer, schistosomula were vortexed for ~2–4 min at 70% vortex power, transferred back to 20 μm mesh and flushed with schistosomula fixative (4% formaldehyde, 0.2% Triton X-100%, 1% NP-40 in PBS) before fixing.

**Imaging and image processing.** Schistosomula FISH images were taken using an Andor Spinning Disk WDb system (Andor Technology). Adult FISH images were taken using a Zeiss LSM 880 with Airyscan (Carl Zeiss) confocal microscope. Colorimetric WISH images were taken using AxioZoom.V16 (Carl Zeiss). Imaris 9.2/9.4 (Bitplane) and Photoshop (Adobe Systems) was used to process acquired images of maximum intensity projections (of z-stacks) and single confocal sections for linear adjustment of brightness and contrast.

**Calculating cell numbers in schistosomula.** Staining schistosomula to visualise cells: Cercariae and parasites at 0, 24 and 48 hr post-transformation were fixed in 5% (v/v) formaldehyde 4% (w/v) sucrose in PBS for 15 min (throughout staining worms were in 1.5 ml microfuge tubes and spun 2 min 500G when exchanging solutions). The parasites were then permeabilised in 10% (w/v) sucrose, 0.5% Triton-X 100 (v/v) for 10 min. Parasites were either stored at 4 °C in 2% formaldehyde in PBS, or stained immediately. Staining was in low light level conditions to minimise photobleaching. For staining, 1 μg/ml DAPI in PBS was added for 10 min, then parasites were post-fixed in 10% formaldehyde in PBS for 2 min. Parasites were washed in 1× PBS then resuspended in 0.4× PBS in ddH₂O (to discourage salt crystals). 10 μl parasites were pipetted onto a glass slide and excess liquid drawn away with whatman filter paper. 10 μl ProLong Gold antifade mountant was added to the sample and a glass coverslip dropped over gently. Slides were left at room temperature overnight to set before imaging. A Zeiss LSM 510 Meta confocal microscope was used in conjunction with the Zen software to take a series of Z stacks, imaging 3 individual worms from each timepoint.

　　Image analysis to calculate cell numbers in schistosomula: Z stack images were imported into ImageJ software (Import>image sequence) then converted to RGB and split by colour (Colour > split channels) and the blue channel used for further processing. Using the metadata associated with the file the scale properties were adjusted. The image was cropped if necessary to show only one parasite. The threshold was set to remove any background. The signal above threshold was measured for the whole image stack (image can be inverted and converted to 8 bit for this purpose). The ROI manager was used to measure individual cell nuclei throughout the Z stack by drawing around the cell on each image of the stack where present. This was imported to the threshold filtered stack and the area measured. 10 nuclei that were clearly defined and of diverse location and size were measured for each worm to obtain an average nuclei size and signal. In all cases as well as X and Y, Z was used to account for the full volume of the nuclei. The total volume for above threshold signal in the worm was divided by the average nuclei size to obtain an estimate for cell number.

**Statistics and reproducibility.** For schistosomula FISH, single FISH was initially performed for each gene and consistency in expression was determined across 5–10 worms. Following single FISH, multiple double FISH experiments were performed with various marker combinations, unless the marker has previously been extensively used in other studies (e.g., *histone h2b* (Smp_108390)[6,47,61], *meg-4* (Smp_307220)[6,59,61], *cathepsin B* (Smp_103610)[6,47,61,66], *tsp-2* (Smp_335630)[6,7,61]). Consistency in expression patterns was determined between 5–10 worms within the experiment, and also across different experiments including single FISH. For schistosomula dextran labelling, two independent experiments were performed with 30-50 worms in each experiment. For adult ISH, in most cases, we first performed WISH with ~5 males and ~5 females and confirmed the consistency in expression across animals, unless the marker has previously been used extensively in other studies (e.g., *histone h2b* (Smp_108390)[6,47,61], *meg-4* (Smp_307220)[6,59,61], *cathepsin B* (Smp_103610)[6,47,61,66], *tsp-2* (Smp_335630)[6,7,61]). Following WISH, we performed multiple FISH experiments to confirm the consistency in expression pattern across FISH experiments. Most double FISH experiments were reciprocated by swapping DIG, DNP or FITC labels for each gene to rule out variations between the labelled probes. The total numbers of independent single and double FISH experiments (including swapped probes) for each gene (schistosomula, adults) are: *hypothetical* (Smp_161510) $N = 3, 7$; *wnt-2* (Smp_167140) $N = 4, 5$; *rhodopsin GPCR* (Smp_153210) $N = 5, 0$; *myoD* (Smp_167400) $N = 3, 5$; *troponin* (Smp_018250) $N = 0, 2$; *actin-2* (Smp_307020) $= 6, 5$; *troponin* (Smp_059170) $N = 0, 4$; *annexin B2* (Smp_077720) $N = 5, 3$; *hypothetical* (Smp_022450) $N = 7, 0$; *meg-3* (Smp_138070) $N = 5, 0$; *ccdc74* (Smp_030010) $N = 4, 2$; *mboat* (Smp_169040) $N = 5, 4$; *dynamin* (Smp_129050) $N = 3, 0$; *epsin-4* (Smp_140330) $N = 3, 5$; *fimbrin* (Smp_037230) $N = 0, 4$; *gtp-4* (Smp_105410) $N = 0, 3$; *lipopolysaccharide induced* (Smp_025370) $N = 0, 1$; *rab18* (Smp_169460) $N = 0, 4$; *NMDA receptor glutamate binding chain* (Smp_181470) $N = 0, 1$; *cathepsin B* (Smp_141610) $N = 5, 3$; *LAP* (Smp_030000) $N = 3, 2$; *serpin* (Smp_090080) $N = 5, 3$; *histone h2a* (Smp_086860) $N = 5, 2$; *cam* (Smp_032950) $N = 3, 5$; *7b2* (Smp_073270) $N = 3, 5$; *gnai* (Smp_246100) $N = 3, 3$; *hypothetical* (Smp_203580) $N = 5, 5$; *Sm-kk7* (Smp_194830) $N = 3, 5$; *fbx* (Smp_132210) $N = 2, 2$; *meg-4* (Smp_307220) $N = 2, 1$; *cathepsin B'* (Smp_103610) $N = 2, 2$; *histone h2b* (Smp_108390) N = 2, 2; *tsp-2* (Smp_335630)

$N = 2, 1$. The complete list of the number of in situ hybridization is shown in Supplementary Data 7.

**Reporting summary.** Further information on research design is available in the Nature Research Reporting Summary linked to this article.

## Data availability
The raw data used in this study has been deposited in ENA with accession number PRJEB34071. The individual sample IDs in ENA are the following: ERS3714216 (FUGI_R_D7119553), ERS3714223 (FUGI_R_D7159524) and ERS3714217 (FUGI_R_D7159525). The data has also been deposited in ArrayExpress with accession number E-MTAB-9684. The data can be visualised and navigated from the following website: https://www.schistosomulacellatlas.org/.

## Code availability
The code used to analyse the data can be found using the following address: https://zenodo.org/badge/latestdoi/271030910[107].

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

## Acknowledgements

Wellcome provided core-funding support to the Wellcome Sanger Institute (Sanger), award number 206194. The work was supported by the Wellcome Strategic Award number 107475/Z/15/Z. P.A.N. is an investigator of the Howard Hughes Medical Institute. *B. glabrata* snails used in the United States were provided by the NIAID Schistosomiasis Resource Center of the Biomedical Research Institute (Rockville, MD) through NIH-NIAID Contract HHSN272201700014I for distribution through BEI Resources. We thank the following individuals at Sanger: Gal Horesh for initial technical assistance optimising dissociation conditions; Catherine McCarthy and Simon Clare for assistance and technical support with animal infections and maintenance of the *Schistosoma mansoni* life cycle; David Goulding and Claire Cormie at the Electron and Advanced Light Microscopy facility; Jennie Graham and Sam Thompson at the Cytometry Core Facility; Nancy Holroyd, Mandy Sanders, Elizabeth Cook and Nathalie Smerdon for facilitating the submission of 10X samples; Matthew Jones for 10X training and library preparations; Cellular Genetics Informatics, especially Martin Prete and Vladimir Kiselev, for creation of a data visualisation website. We thank Dr. Shristi Pandey for sharing the random forest code used in this work and Dr. Mireya Plass for sharing the planaria dataset. Finally, we thank the single cell online community for enthusiastically sharing their work online.

## Author contributions

M.B., G.R., P.A.N., J.L. and C.L.D.S. conceived of the experiments. C.L.D.S. analysed the data, with contributions from A.C., A.T., M.D.Y., T.A., Z.L., H.M.B and J.L. Validation/ experiments were performed by J.L. with contributions from T.Ch., K.R. and C.L.D.S. Initial data preparation was performed by S.R.D. and S.L. FACS sorting and analysis was performed by C.H. and B.L.N. The paper was written by C.L.D.S., J.L., M.B. and G.R. The paper was edited by C.L.D.S., J.L., M.B., G.R. and P.A.N. The study was co-directed by M.B., G.R. and P.A.N.

## Competing interests

H.M. Bennett is currently employed at Berkeley Lights Inc. which makes commercially available single-cell technology
