## [Peer Review File · Nature Communications]

Reviewers' Comments:

Reviewer #1:

Remarks to the Author:

This is an interesting paper that delineates the distinct signatures present in different single cell clusters that represent most of the tissues from 2-day old schistosomula, the early intra-mammalian life-cycle stage of the *Schistosoma mansoni* parasite. The data presented here give insight into three different subtypes of neural cells, which have otherwise been almost impossible to identify in bulk cellular and transcriptomic studies. It also shows two populations of parenchyma cells, which represent a tissue that has been less characterized so far. Although this is a descriptive study, it provides an enormous amount of information that will be useful in the future to gain mechanistic insights into the developmental biology of the parasite.

1) The Results start by the authors describing that they first developed a protocol to efficiently dissociate the parasites using a protease cocktail (line 85). Going to the Methods to check the protocol (see line 376) I see that the authors mention the use of Liberase DL without describing the medium that was used. Was it Basch's medium, or another medium? Please specify the exact composition of the dissociation medium. Also note that in the Methods (line 376) 750 ug/ml Liberase DL is stated; however, in Supplementary Figure 1 an amount of 750 ul/ml is shown. Please indicate which is the correct one.

2) On lines 89-90 the authors describe that they detected "a median of 900 genes per cell and depth of 283,000 reads per cell (Table S1)". Please mention in the text the number of UMIs per cell in addition to reads/cell, and also give the corresponding information in Table S1 for each sample replicate. I wonder why such a small number of genes was detected per cell with such a deep sequencing coverage per cell. In the comparative analysis of single-cell RNA-Seq sensitivity among the different systems, Zhang X. et al. (Molecular Cell 73: 130-142.e5, 2019) have recently shown that 10X Genomics system is expected to detect between 3,000 and 3,500 genes when having a median reads per cell higher than 50,000. Can the authors raise some possible hypothesis for having achieved such a low sensitivity? I wonder if some bias caused by the read-mapping tool used and its corresponding mapping parameters has systematically eliminated a certain set of genes with any particular characteristic (number and size of exons, eventual low complexity strings of bases in the cds, etc.). This is because I noticed that only "approximately 55% of sequenced reads mapped confidently to the transcriptome" (line 407) "using the 10X Genomics analysis pipeline Cell Ranger (v 2.1.0)" (lines 405-406). Did the authors try any other mapping tool such as STAR and tested if a higher fraction of their sequenced reads were mapped? If yes, did the authors try and see if a higher number of genes could be detected, indicating a systematic loss of some type(s) of genes?

3) Also, still regarding the analyses pipeline I got somewhat confused with the effective clustering approach. Thus, on lines 94-96 the authors state: "[...] we used a combination of the SC3, Seurat and UMAP algorithms to cluster cells based on their mRNA expression levels [...]". How were these 3 tools combined? Upon inspecting Tables S2-S5 that show the different clustering results, I was further confused with two different Seurat gene lists (Tables S4 and S5). Table S4 has the label "All Seurat markers_toc" and 1,033 marker genes, whereas Table S5 has "All Seurat_Wilcoxon" and 5,646 marker genes. There is no explanation for the meaning of "All Seurat markers_toc" and for the difference in the analyses. Table S3 is labeled "All SC3 markers" and has 10,425 marker genes. Again, with such a widely different number of marker genes, how was the combination of tools achieved? Also, on lines 101 to 105 the authors state: "Based on the marker genes identified using Seurat, we identified [...]" followed by a list of all cell clusters and the corresponding numbers of cells in each cluster. It seems to me that essentially the Seurat tool was used. Please clarify the clustering approach in the Results and in the Methods.

3) The GO analysis of the marker genes (lines 106-107) of all clusters (Supplementary Figure 1C) is

interesting, and it shows that generally the GOs matched the predicted cellular processes for each cluster. However, I feel the need of a heatmap of the differentially expressed marker genes that belong to the enriched GO categories of each cluster. The heatmap would give an immediate visual appraisal of the clusters in which the markers are indeed most highly expressed, as opposed to other clusters that are not as well discriminated, where the supposed specific markers are shared by two or three clusters. A good example of a heatmap connected to GO clusters can be found in Fig. 2 of Farbehi et al. eLife 2019; 8:e43882 (<https://doi.org/10.7554/eLife.43882.013>).

4) On line 116 the authors describe: "Three discrete muscle clusters were identified by examining the expression of [...]". Please note that only the total number of cells identified in the three muscle clusters together is given in line 102. How many cells were identified in each of the three muscle clusters? Please cite these numbers.

In fact, when describing each tissue throughout the paper, for all those tissues where multiple clusters were identified, the authors should give the separate numbers of cells identified in each of the different clusters within a tissue type.

5) Related to the three different muscle clusters, I have a problem with the "second muscle-like cluster" (lines 127-131), named myoD+ muscle. Specifically, I am very much concerned with the fact that most of the rhodopsin GPCR (Fig 2E) is NOT co-localized with actin-2 by FISH. On one hand, from Figure 2A one can see that rhodopsin GPCR gene is expressed in cells belonging to Sm-kk7+ neurons almost as much as in the myoD+ muscle. The image of Fig 2E is consistent with rhodopsin GPCR gene marking other cell types away from actin-2-expressing muscle. On the other hand, Fig 2A shows that only approx. 50 % of cells in the myoD cluster express myoD, and only approx. 50 % of cells in this cluster express rhodopsin GPCR. Given the very low number of double-positive stained cells in FISH and the partial frequency of cells expressing either of the two genes in the RNA-seq data, I am not convinced that the so-called "myoD+ cluster" is a homogeneous true cluster. In fact, Fig 1B shows that this is the most disperse cluster and perhaps the least dense of the three muscle clusters (how many cells comprise this cluster?)

Here, I believe that it is necessary that the authors modify Fig. 2A and accompany it with two additional t-SNE representations of clustered cells (similar to Fig 1B) all painted in gray, and where in one of them the cells expressing the rhodopsin GPCR marker gene are colored in red and in the other the cells expressing myoD are colored in red. In this way, one can see what is the fraction of cells that were detected as co-expressing those supposed two markers of the "myoD+ cluster" and see the fraction of cells that separately express either one of the two genes.

Again, the paper previously cited (Farbehi et al. eLife 2019; 8:e43882, <https://doi.org/10.7554/eLife.43882.013>) is one of many papers in the literature that show the expression of each cluster marker gene in a t-SNE plot with gray-colored background, in the way that I suggested above.

6) In fact, I think that all dot plot figures from each of the clusters (from Fig. 2A through Fig. 6A) should be accompanied by separate t-SNE gray-colored plots, where the cells that express each of the two or three marker genes (selected as markers from RNA-Seq data) are highlighted in red in a separate t-SNE plot for each marker gene. In this way, one can see that the marker genes were indeed detected as simultaneously expressed in the same population of cells in the cluster.

7) On line 129: "Both genes showed a scattered expression pattern throughout the schistosomula (Figure 2E), [...]". This sentence refers to both rhodopsin GPCR and myoD being expressed throughout the schistosomula; however, myoD FISH data was NOT shown anywhere in schistosomula, only in adults (Fig 2F). Please correct the text in the Results and the scheme in Fig 2H.

8) Lines 159-160: "We found that cells expressing meg17 also expressed the known oesophageal marker meg4 (Ref. 62) [...]". Reference 62, namely Wilson et al. (2015) is cited for MEG4 known

oesophageal marker. Please note that although Wilson et al. (2015) show that MEG4 gene is among the 27 MEGs that were detected by RNA-seq to be enriched in a homogenate preparation of adult male heads compared with a homogenate of isolated adult tails, that paper does not show MEG4 as an esophageal marker. The paper by DeMarco et al., 2010, *Genome Research* 20: 1112-1121 shows by whole mount in situ hybridization that MEG4 is expressed in the primordial esophagus of day 3 schistosomula. This work should be cited instead. Also, in the scheme of Fig 3H the paper of Dillon et al. 2007 is cited for MEG4 in the esophageal gland, and it should be replaced by DeMarco et al., 2010 (no MEGs were probed in the Dillon et al. 2007 paper, only protein-coding genes with known functions).

9) On lines 162-163 the authors stated: "Distinguishing the second tegumental cluster was challenging due a lack of Tegument 2-specific markers (Figure 3A)." I noticed in Figure 3A that mboat gene (Smp_169040) was enriched in Tegument 2 as opposed to Tegument 1 and it could have been used as a distinguishing marker. Mboat belongs to the family of Membrane Bound O-Acyltransferase proteins, an enzyme that acylates membrane lysophospholipids. This enzyme could eventually be related to the acylation of specific phosphatidylinositol membrane phospholipids that may be required for clathrin-related endocytosis. The next paragraph (lines 171-179) discusses components of the clathrin pathway identified in Tegument 2; this enzyme could be cited as possibly related to the clathrin pathway.

10) In Figure 4F Caffrey et al. 2004 is cited for cathepsin B' as a known marker for parenchyma. Please note that the paper is not in the references list. Caffrey et al. *Trends Parasitol.*, 20 (2004), pp. 241-248 must be cited.

11) I have already mentioned before, and I will repeat myself here. How many cells were detected in each of the three neural-associated clusters (line 238)? 7b2+ cells appear to be more abundant than the other two types of neural cells, according to the ISH images. Does the number of cells detected in the 7b2+ cluster reflect the apparent higher number of 7b2 positive cells in schistosomula? Please comment in the text.

12) Lines 254-255: "[...] marker gene encoding KK7 (Smp_194830), known to be associated with the peripheral nervous system in *S. mansoni* (Ref. 55). Please note that Ref 55 is badly formatted in the references list. The author's name is misspelled. It should read "Manuel, S. J." not "SJ, M."

13) The use of random forest (RF) machine learning (line 270) to compare *S. mansoni* with *Schmidtea mediterranea* clusters obtained with single-cell RNA-seq is an interesting approach. However, the Methods related to RF (lines 508-520) are confusing. Specifically, it is not clear if the evaluation of the RF classifier on the *Schmidtea* dataset (line 508) was performed with only the 692 orthologous genes between *S. mansoni* and *S. mediterranea* mentioned in lines 519-520. Since the RF classifier used for the *Schistosoma* dataset contained only those orthologous genes, it seems to me that the initial evaluation of the RF classifier on the *Schmidtea* dataset should also be done with only those 692 genes. Please clarify.

14) On lines 330-331 the authors discuss about the challenges that they faced, pointing that in spite of the challenges they successfully characterised several previously unknown marker genes and populations. I would add another difficulty possibly related to the fact that an incomplete reference transcriptome was used. It has already been shown that *S. mansoni* expresses thousands of long non-coding RNAs (lncRNAs) (Vasconcelos et al. 2017, *Sci Rep.* 7: 10508; Liao et al. 2018, *Exp. Parasitol.* 191: 82-87) and that a few hundred lncRNAs are specifically expressed in schistosomula (Maciel et al. 2019, *Front. Genet.* 10: 823). Also, some lncRNAs are gene markers of different cell populations of single-cell juvenile and mother sporocysts stem cells (Maciel et al. 2019, *Front. Genet.* 10: 823). It is possible that lncRNAs are also gene markers for the different tissues that were isolated and characterized here. A future re-analysis of the single-cell RNA-Seq data obtained here that includes the lncRNAs might help to uncover some cell types that were not identified by only looking at the

protein-coding genes. I think that this aspect is missing from the Discussion.

Reviewer #2:

Remarks to the Author:

Authors used scRNAseq to characterise 2-day schistosomula transformed mechanically by using 10X Chromium technology, followed by validation of the cell clusters by RNA in situ hybridization (ISH) in schistosomula and adult worms. This study resulted in 11 discrete cell populations identified and novel marker genes validated for muscles, nervous system, tegument, parenchymal/gut primordia and stem cells. This study enhance a better understanding of cell types and tissue differentiation in schistosomula and is essential for unravelling the developmental biology of this important parasite. However, I think it would have been much stronger if the authors would have used 14-28 day old juvenile worms from mice and not simply cultured schistosomula (2-day old, which have different biochemical parameters from the in vivo grown worms). The authors could have also validated their findings, running a side by side experiment using ex vivo worms as well as schistosomula to show behave similarly.

Previous bulk RNA-seq studies have demonstrated that ~11,000 genes presented in *S. mansoni* schistosomula (PLoS Negl Trop Dis 7(3): e2091), however, in this study, by using single cell sequencing technology 900 genes per cell (from 2,144 cells) were identified and some typical cells were not successfully isolated from schistosomula cells, my concerns are:

- 1) Is Liberase DL suitable for isolation schistosome cells? Results showed total 33 genes were identified from schistosomula tegument cells, excluding a number of typical tegument genes (such as TSP-2, triose-phosphate dehydrogenase) which have been demonstrated the surface location in the parasites. That indicates that Liberase DL may cause damage of tegument cells or other parasite cells.
- 2) The stress of schistosomula cells induced by the processing of generating alive single-cell suspension including enzymatic digestion of schistosomula and con-staining with FDA and PI and cell sorting, may result in the changes of gene profiles of cells.
- 3) In this study, approximately 7000 cells per reaction were supposed to be loaded for single cell sequencing, actually, only about 1200 cells per experiment were captured. That may lead to lose information from uncaptured cells, which brings difficult to obtain RNA information from rare cell populations or from rare genes with low gene copy number. Can you explain why there is only limited cells were captured for the analysis? Do authors have any ideas to improve the protocol? It is necessary to optimize current method due to in-depth transcriptome analysis requires the profiling of a large number of cells.
- 4) As known, read counts observed are affected by a combination of different factors, including biological variables and technical noise. Critically, the small amount of starting material used in scRNA-seq may amplify the effects of technical noise. However, given the high cost of scRNA-seq and the lack of standards in regard to analyzing it, different experiments (other than ISH) to bolster the main conclusions are needed.
- 5) Can you give details about library preparation for the scRNA-seq?

Reviewer #3:

Remarks to the Author:

In this work, Soria et al. present the first complete atlas of the schistosomulum, the first intra-mammalian development stage of the parasite *Schistosoma mansoni*. The authors use an in vitro system to obtain schistosomula and generate an atlas containing more than 2000 cells using 10X genomics sequencing technology. In this atlas, they identify 12 clusters, 11 of them representing transcriptionally different cell types, which they validate in schistosomula and adults using ISHs. Additionally, they compare the clusters obtained to those from *S. mediterranea*, the closest free living relative from *S. mansoni*, in order to identify similarities in the cell repertoire of *S. mansoni* schistosomulum. The work is interesting and sound, although there are some concerns that prevent

me from recommending the publication of this article in its current format.

Major Comments:

- The authors mention in the text that they use SC3, Seurat and UMAP to cluster the data. However, in the methods section they just describe independent filtering and clustering analyses performed with Scater package, SC3 and Seurat, but they do not describe how these methods have been integrated, nor how they used UMAP for Clustering. Considering that clustering is one of the key steps in single-cell transcriptomics analyses, I would ask the authors to provide additional information about how the clustering has been performed. For instance, they should provide information about how they selected the genes used for clustering, the selection of PCAs (if applicable) for each method, and the integration of the methods to generate the final set of clusters.
- The authors describe an ambiguous cluster that represents around ~10% cells obtained. They claim that it could not be “experimentally defined”. Yet, the authors do not describe which are the methods that have been used to characterize this cell population nor include it in any of the future analyses comparing the different clusters included in the main manuscript nor in the supplementary materials. According to the markers identified by the authors (Tables S2-S6), the ambiguous cluster expressed specific marker genes that could be used to validate this cell population using ISH. Thus, the authors should include the results of the experimental validation of this cell cluster as well as in the rest of the plots in the manuscript (Fig1-7) in order to understand the relation of this cell population to all the other populations identified. This will be necessary to understand if this is a new cell population or rather low quality cells, doublets or other artifacts.
- The authors define a positional muscle cluster based on the expression of the uncharacterized gene *Smp_161510* and *wnt* (*Smp_167140*). However, according to the data provided, there is no coexpression of these markers in the schistosomulum (Figure 2B,C), nor in adult worms (Figure 2D, Supplementary Figure 2). Thus, it seems that this *wnt*+ population may be a distinct population of cells that the authors have not been able to capture in their clustering. The authors should provide tSNE plots showing the expression of these two genes in the single-cell dataset as well as co-stainings showing the co-occurrence of the two markers.
- The authors should explain how they define the set of marker genes shown in Figure 3A as well as in the rest of panel A for all main Figures. According to table S4, Tegument 2 cell type has significant marker genes not present in tegument 1 cell such as *Smp_0745701* and *Smp_1694601*. Besides, the authors also show tegument 2 specific genes in Supplementary Figure 3H & F. The authors should provide validation of these markers or other exclusive markers from tegument 2 in order to characterize the differences across the two cell populations.
- The authors should make a unified set of marker genes instead of reporting three independent sets of marker genes (Tables S3-S5).
- The authors used a Random Forest classifier to identify similarities across schistosomula and *S. mediterranea*. The approach is interesting but the authors should explain the performance of the method. For instance, they report that they evaluate the classifier using the Schmidtea dataset but they do not report the performance of the method, i.e. the accuracy of the classification. Additionally, the authors should explain how they annotated the clusters from *S. mediterranea* given that the original publication contained a different number of clusters. The authors show in Fig. 7 that most of their identified cell types correspond to “*psd*+ cells: *neoblast1*” identified in *S. mediterranea*. *Psd*+ cells is a rather small cluster in the original publication whereas *neoblast1* is the main stem cell group identified. The authors need to reinterpret the correspondence of clusters as it seems that most of them have stem cell related genes rather than *psd*+ cells genes.
- In the discussion, the authors acknowledge the possibility of missing known cell populations because they cannot identify a specific cluster containing them, such as the case of the protonephridia cells. To

support their claims, the authors should provide some plots showing the expression of known protonephridia marker genes in the tSNE plot.

Minor Comments:

- The authors report the number of reads per cell. This number is not relevant given that it could reflect only sequencing of PCR duplicates. The authors should report instead the number of UMIs per cell.
- Why do the authors use characterized genes in human and mouse to define cell populations and not those from closer relative species?
- Table S3 lacks the association of ~3000 genes to identified clusters
- The authors should provide additional details about how they define top markers (included in table S2) as well as which are the parameters used to define markers with Seurat.
- Figure 2A in line 134 should be Figure 2G.
- I cannot find the ISH pictures of the validation of fimbrin in Fig 2 or Supp. Fig 2.
- The authors should define if the ISH images report dorsal or ventral views.
- The authors should provide tSNE plots showing the expression of all the validated marker genes.
- The zoomed area in Figure 3G does not correspond to the region highlighted in the general FISH picture on the left.
- I do not see coexpression of Smp_022450 and annexin B2 in schistosomulum heads.
- The ISH showing the expression of Smp_022450 in Figure 3B is very different from the staining obtained in Figure 3D, 3F and 3G, in particular in relation to the expression of the gene in the body. The authors should explain this discrepancy.
- In relation to the analysis performed by the authors in Suppl. Figure 3, does Tegument 1 also have specific functions and thus tegument 1 and 2 represent two functionally distinct cell types of the tegument? The authors should clarify this point.
- Rename the panels in Suppl Figure 4 so that they are cited in the text in the same order as in the main text.
- Include the cell type for which Smp_022450 is a marker in Figure S4G.
- Remove "that" in line 200. "genes mark schistosomula parenchyma,".
- Rename panels in Figure 5 so that they appear in the same order as in the text.
- Authors claim that cam+ cells are h2b+ (line 222). Yet, in figure 5 they show little overlap. Please rephrase.
- The authors should describe in detail how did they make the plots shown in Figure 5C and which are the previously described stem cell populations depicted there.
- Mark the location of the main and minor nerve cords in Figure 6G and Supplementary Figures 6E & F.

- There is a typo in Figure 6H. It says "gnia" instead of "gnai"
- In the Random Forest method description, the authors have a discrepancy in the number of cells used: from 21610 they keep 21612 cells.

Reviewer #1 (Remarks to the Author):

This is an interesting paper that delineates the distinct signatures present in different single cell clusters that represent most of the tissues from 2-day old schistosomula, the early intra-mammalian life-cycle stage of the *Schistosoma mansoni* parasite. The data presented here give insight into three different subtypes of neural cells, which have otherwise been almost impossible to identify in bulk cellular and transcriptomic studies. It also shows two populations of parenchyma cells, which represent a tissue that has been less characterized so far. Although this is a descriptive study, it provides an enormous amount of information that will be useful in the future to gain mechanistic insights into the developmental biology of the parasite.

We would like to thank the referee for such positive comments.

1) The Results start by the authors describing that they first developed a protocol to efficiently dissociate the parasites using a protease cocktail (line 85). Going to the Methods to check the protocol (see line 376) I see that the authors mention the use of Liberase DL without describing the medium that was used. Was it Basch's medium, or another medium? Please specify the exact composition of the dissociation medium. Also note that in the Methods (line 376) 750 ug/ml Liberase DL is stated; however, in Supplementary Figure 1 an amount of 750 ul/ml is shown. Please indicate which is the correct one.

The dissociation medium was: 750µg/ml Liberase DL in 1X PBS supplemented with 20% FBS. The manuscript (Methods and Supplementary Figure legend) has been corrected accordingly.

2) On lines 89-90 the authors describe that they detected “a median of 900 genes per cell and depth of 283,000 reads per cell (Table S1)”. Please mention in the text the number of UMIs per cell in addition to reads/cell, and also give the corresponding information in Table S1 for each sample replicate.

We have updated the Methods section and the Supplementary Table 1 with the number of UMIs per cell as suggested by the reviewer. For the reviewer's reference, we have included screenshots of the different QC metrics provided by CellRanger software for our samples (see “Additional Information 1”, at the end of this document)

I wonder why such a small number of genes was detected per cell with such a deep sequencing coverage per cell. In the comparative analysis of single-cell RNA-Seq sensitivity among the different systems, Zhang X. et al. (Molecular Cell 73: 130-142.e5, 2019) have recently shown that 10X Genomics system is expected to detect between 3,000 and 3,500 genes when having a median reads per cell higher than 50,000. Can the authors raise some possible hypothesis for having achieved such a low sensitivity? I wonder if some bias caused by the read-mapping tool used and its corresponding mapping parameters has systematically eliminated a certain set of genes with any particular characteristic (number and size of exons, eventual low complexity strings of bases in the cds, etc.). This is because I noticed that only “approximately 55% of sequenced reads mapped confidently to the transcriptome” (line 407) “using the 10X Genomics analysis pipeline Cell Ranger (v 2.1.0)” (lines 405-406). Did the authors try any other mapping tool such as STAR and tested if a higher fraction of their sequenced reads were mapped? If yes, did the authors try and see if a higher number of genes could be detected, indicating a systematic loss of some type(s) of genes?

We apologise as there was an error in the reported number of reads mapping confidently to the transcriptome in our manuscript. Specifically, the correct values for reads that confidently mapped to the genome are 73.1% (sample FUGI_R_D7119553), 63.0% (FUGI_R_D7159524), and 64.3% (FUGI_R_D7159525). Accordingly, we have now corrected the manuscript to state that approximately 67% of reads mapped to the genome (line 431). For the reviewer’s reference, we show the output statistics from 10X Cell Ranger Software in Additional Information 1.

We used the default options in CellRanger, which uses STAR as a mapping tool. The metrics for ‘number of reads mapping confidently to the transcriptome’ is within range seen for other samples prepared using the same version of 10X Chromium reagents. For example, across 21 epithelial samples from mice prepared using the same version (v2) of 10X Chromium, our group has found the ‘number of reads mapping confidently to the transcriptome’ to be 68%–75%.

The paper highlighted by the reviewer from Zhang X. et al. (PMID:30472192) benchmarked three different droplet technologies using a lymphoblastoid cell line GM12891. This represents an ideal case scenario and the numbers are not representative of experiments involving tissue dissociation. As a more representative example, we compared our statistics to a larger sample of single cells (125,141 cells) from dissociated kidney tumours by Young et al., 2018 (PMID:30093597). We found that the number of genes per cell for their dissociated tissue was similar to ours (898 genes per cell). The data for Young et al., 2018 experiment corresponds to TableS11 and can be accessed via: https://science.sciencemag.org/highwire/filestream/713964/field_highwire_adjunct_files/5/aat1699-Young-TablesS1-S12-revision2.xlsx

Furthermore, the total number of genes varies between organisms and so comparisons between humans and Schistosoma must be adjusted for total gene number. We captured 900 genes/cell out of a total of 10,129, or 8.9% of the possible genes. This represents a much higher rate of capture than Young et al., who captured only 898 out of a possible 33,645 genes, or 2.7% of the total. If we take the average of the total of genes detected across the single-cell experiments (Supplementary Table 1), we find expression of 8,759

genes. This represents 86% of the total number of protein coding genes (10,129 genes) found in the *S. mansoni* genome reference version 7.1.

Finally, we found that when aggregated at the cluster level, we capture a very high fraction of *S. mansoni* genes. For example, although on average each cell in the myoD+ cluster expressed only 900 genes, when the expression was aggregated we found 4074 genes (raw counts >50) and 7084 genes (raw counts > 0) are expressed.

3) Also, still regarding the analyses pipeline I got somewhat confused with the effective clustering approach. Thus, on lines 94-96 the authors state: “[...] we used a combination of the SC3, Seurat and UMAP algorithms to cluster cells based on their mRNA expression levels [...]”. How were these 3 tools combined?

We agree with the referee that this information was not clear. We made extensive changes to the Methods section “Quality Control of single-cell data”. In addition, we changed the following sentence in the Results section.

In our analysis, we used SC3 (Kiselev et al. 2017, PMID:28346451) to filter out low-quality cells. We have edited the Methods section to state that SC3 was used purely as a QC tool.

from:

“To create a cellular map of the *S. mansoni* schistosomula, we used a combination of the SC3, Seurat and UMAP algorithms to cluster cells based on their mRNA expression levels and statistically identify marker genes that were best able to discriminate between the clusters”.

to:

“To create a cellular map of the *S. mansoni* schistosomula, we used Seurat⁴² to cluster and identify marker genes that were best able to discriminate between populations...”

We mentioned UMAP in the manuscript because we constructed a UMAP using scanpy package. However, this was done in addition to the main analysis to assess reproducibility of results across different single-cell analysis packages. This extra analysis was not used to pick gene markers or to assign populations. We originally included this information for completeness but, on reflection, it does not contribute anything to the manuscript. We have removed it from the revised version.

SC3 was used purely as a QC tool to exclude low quality cells. In the first instance we used a range of values close to the k value estimated using the `sc3_estimate_k` function (k=26) from the SC3 package (Additional Information 2A). In other words, SC3 estimated 26 clusters in our dataset. The stability and quality of the clusters was assessed by inspecting the data obtained for the specified k value ranges. Clusters with stability index less than 0.10 and/ or less than 3 cells were excluded from further analysis. We continued to re-cluster cells until all clusters had stability values greater than 0.6 and contained more than 5 cells. This allowed us to keep cells that could be confidently assigned to a specific cluster represented by the red blocks (Additional Information 2B). We fed this gene count matrix to Seurat to continue the analysis. For consistency, in order to compare our single-cell dataset to the *S. mediterranea* dataset using the Random Forest analysis we employed the same approach used for *S. mediterranea* (Seurat).

Additional Information 2. Consensus metrics generated by SC3. (A) Consensus metrics of the raw dataset (before QC) comprising 3,513 larval cells. (B) Consensus metrics of 2,144 cells after SC3 QC. In both cases the consensus metrics indicate how often each pair of cells is assigned to the same cluster (1 - always; 0 - never).

Upon inspecting Tables S2-S5 that show the different clustering results, I was further confused with two different Seurat gene lists (Tables S4 and S5). Table S4 has the label “All Seurat markers_toc” and 1,033 marker genes, whereas Table S5 has “All Seurat_Wilcoxon” and 5,646 marker genes. There is no explanation for the meaning of “All Seurat markers_toc” and for the difference in the analyses. Table S3 is labelled “All SC3 markers” and has 10,425 marker genes. Again, with such a widely different number of marker genes, how was the combination of tools achieved? Also, on lines 101 to 105 the authors state: “Based on the marker genes identified using Seurat, we identified [...]” followed by a list of all cell clusters and the corresponding numbers of cells in each cluster. It seems to me that essentially the Seurat tool was used. Please clarify the clustering approach in the Results and in the Methods.

We agree with the reviewer that this was unclear. As explained above, we used SC3 as QC only. Providing all these data was confusing and counterproductive. We have therefore removed the information on the SC3 markers because they were not subsequently used (whereas information on the markers predicted by Seurat has been retained because these were the lists used to provide validation candidates). Seurat provides multiple options to calculate marker genes. For completeness, we included lists for two of these tests (ROC and Wilcox). However, we agree with the reviewer that this was confusing, and have simplified the manuscript by only retaining the list of marker genes calculated using the ROC test. The latter provides a confidence (AUC) value on how discriminatory each marker should be for

marking a particular cluster. This was key information that we used when manually selecting marker genes for validation.

4) The GO analysis of the marker genes (lines 106-107) of all clusters (Supplementary Figure 1C) is interesting, and it shows that generally the GOs matched the predicted cellular processes for each cluster. However, I feel the need of a heatmap of the differentially expressed marker genes that belong to the enriched GO categories of each cluster. The heatmap would give an immediate visual appraisal of the clusters in which the markers are indeed most highly expressed, as opposed to other clusters that are not as well discriminated, where the supposed specific markers are shared by two or three clusters. A good example of a heatmap connected to GO clusters can be found in Figure 2 of Farbehi et al. eLife 2019; 8:e43882 (<https://doi.org/10.7554/eLife.43882.013> [doi.org]).

We agree with the reviewer that a heatmap provides a clear representation of the GO enrichment analysis for each of the cell clusters. We performed a GO term enrichment on the list of 490 genes with AUC > 0.7 and constructed a heatmap as suggested. We have now added this figure as Supplementary Fig. 2.

5) On line 116 the authors describe: “Three discrete muscle clusters were identified by examining the expression of [...]”. Please note that only the total number of cells identified in the three muscle clusters together is given in line 102. How many cells were identified in each of the three muscle clusters?

In fact, when describing each tissue throughout the paper, for all those tissues where multiple clusters were identified, the authors should give the separate numbers of cells identified in each of the different clusters within a tissue type.

This is a good suggestion and accordingly, the number of cells for each cluster is now shown throughout.

5) Related to the three different muscle clusters, I have a problem with the “second muscle-like cluster” (lines 127-131), named *myoD*+ muscle. Specifically, I am very much concerned with the fact that most of the rhodopsin GPCR (Fig 2E) is NOT co-localized with *actin-2* by FISH.

We’re not 100% confident that we understand the reviewer’s comment/concern. The second muscle cluster is distinguished by the fact that it expresses *myoD* and rhodopsin GPCR but not *actin-2*. We believe this is already clear from Fig. 2a. High *actin-2* is a defining feature of the third muscle cluster, along with very low rhodopsin GPCR expression. The demonstration by FISH that most of the rhodopsin GPCR is NOT co-localised with *actin-2* (now moved to Supplementary Fig. 4f) is therefore entirely consistent with our expectations.

To improve clarity, we have modified the text describing the third as follows:

“Finally, a third cluster (179 cells) of putative muscle cells was distinguished from the other clusters by its high *actin-2* (*Smp_307020*, *Smp_307010*) expression and lower expression of *myoD*, *Smp_161510* and *rhodopsin GPCR* (Fig. 2a and Supplementary Fig. 3a).”

On one hand, from Figure 2A one can see that rhodopsin GPCR gene is expressed in cells belonging to Sm-kk7+ neurons almost as much as in the myoD+ muscle. The image of Fig 2E is consistent with rhodopsin GPCR gene marking other cell types away from actin-2-expressing muscle.

We agree with the reviewer that the text perhaps gave the impression that rhodopsin GPCR was expressed exclusively in the second muscle cluster (myoD+ cluster). Although rhodopsin rhGPCR showed striking expression in that cluster (see Fig. 2a), it is clearly expressed elsewhere. In Fig. 2e (now in Supplementary Fig. 4f) we do indeed show that rhGPCR marks cells that are not actin-2 expressing muscle. To some degree, actin-2 is expressed in the second muscle cluster (myoD+ cluster) but shows much greater expression in the third cluster (“actin 2+ muscle”; Fig. 2a)

We’ve clarified that part of the text to make it clear that rhodopsin GPCR is not uniquely expressed in the second muscle cluster (myoD+ cluster) :

“In a second muscle-like cluster (561 cells), an orthologue (Smp_167400) of the myoD transcription factor from *S. mediterranea* (dd_Smed_v6_12634_0_1)³¹ was uniquely expressed (Fig. 2a). In addition, expression of *rhodopsin GPCR* (Smp_153210) was enriched in this *myoD*+ cluster (Fig. 2a and Supplementary Fig. 3a).”

We also agree that Fig. 2a clearly shows rhodopsin GPCR expression from both the myoD+ cluster and the Sm-kk7+ cluster. However, the size and colour intensity of the dotplots show that the expression values are different. Looking at a table of the underlying values for the figure (included below for information), there is a 5-fold difference in rhGPCR expression levels between these clusters:

Table of average rhGPCR and myoD gene expression values per cell for each cell cluster (normalised counts from Seurat).

	myoD+ Muscle	Posit. Muscle	Actin 2+	7B2/ PC2+ Neurons	Gnai+ Neurons	Sm-kk7+ Neurons	Teg 1	Teg 2	Paren. 1	Paren. 2	Stem/ Germ.
Smp_153210 (rhod. GPCR)	18.7	6.1	3.5	2.6	1.1	3.8	0.4	0.5	0.4	1.6	0.5
Smp_167400 MyoD	9.1	0.2	3.5	0.3	0.3	0	0.3	0.1	0.1	0.4	0.2

In addition, rhGPCR is expressed in 84.7% (475/561) cells in the myoD+ cluster and in 75% (15/20) cells in the Sm-kk7+ cluster.

On the other hand, Fig 2A shows that only approx. 50 % of cells in the myoD cluster express myoD, and only approx. 50 % of cells in this cluster express rhodopsin GPCR. Given the very low number of double-positive stained cells in FISH and the partial frequency of cells expressing either of the two genes in the RNA-seq data, I am not convinced that the so-called "myoD+ cluster" is a homogeneous true cluster.

We thank the reviewer for highlighting this point and apologise that there was in fact a formatting error here that affected interpretation and made the figure misleading. After re-inspecting the figures, we now realise the key for the Dotplot was pasted into position but was not resized correctly. The relativities are of course preserved but it gave the impression that the proportion of cells per cluster expressing each marker was lower. The data in the Dotplot itself were correctly drawn but the incorrectly scaled key gave the impression that ~50% cells expressed myoD, whereas the true proportion is 74.5% (Fig. 2a). The proportion that expressed rhGPCR was 84.7%. We have now rebuilt all of the Dotplots and equally scaled the plot and the key into each figure, but would like to stress that the underlying expression data remain unchanged (Fig. 2-6). We have also provided the t-SNE as requested by the reviewer to demonstrate the expression of myoD in the myoD+ cluster (Supplementary Fig. 3a).

The reviewer refers to a low number of double positive cells but it is not clear to what they are referring. To improve the clarity of the section on muscle, we have now included the result from a FISH experiment probed with myoD in schistosomula (Supplementary Fig. 4c) using hybridization chain reaction (HCR) (see Methods for details). The reviewer also questioned whether the cells that we've labelled myoD+ do indeed comprise a separate homogeneous cluster. Hopefully our response to the preceding comments has made the evidence clearer. We have added some additional sentences in the discussion to emphasise the heterogeneous nature of single cell data:

“...A goal of single cell sequencing is to capture the heterogeneity of all cells and classify them into their broad types. Stochastic gene expression fluctuations mean that cells of the same cluster do not necessarily display the same expression profiles. Our transcriptome analysis enabled the conservative characterisation of 11 distinct clusters...”

In fact, Fig 1B shows that this is the most dispersed cluster and perhaps the least dense of the three muscle clusters (how many cells comprise this cluster?)

The myoD+ cluster comprises 561 cells. Although t-SNE is an excellent technique for data visualisation, it may lose some of the global data structure (Kobak, et al., 2019, PMID:31780648). Therefore, t-SNE should be used solely as a visualisation aid rather than for defining the clusters. To this end, we have included t-SNE plots of marker genes for each cluster in Supplementary Fig. 3.

Here, I believe that it is necessary that the authors modify Fig. 2A and accompany it with two additional t-SNE representations of clustered cells (similar to Fig 1B) all painted in grey, and where in one of them the cells expressing the rhodopsin GPCR marker gene are coloured in red and in the other the cells expressing myoD are colored in red. In this way, one can see what is the fraction of cells that were detected as co-expressing those supposed two markers of the "myoD+ cluster" and see the fraction of cells that separately express either one of the two genes

Again, the paper previously cited (Farbehi et al. eLife 2019; 8:e43882, <https://doi.org/10.7554/eLife.43882.013> [doi.org]) is one of many papers in the literature that show the expression of each cluster marker gene in a t-SNE plot with gray-coloured background, in the way that I suggested above.

We thank the reviewer for this suggestion and have created grey/red feature plots to show marker gene expression. Because many of the multipanel figures are already rather crowded, incorporating the grey/red figure made the layout look quite confusing, so we have elected to collect these supporting figures together in a new Supplementary Fig. 3. For myoD and rhGPCR and the differential distribution of rhGPCR and rhGPCR+myoD+ expressing cells can now be seen more clearly (Supplementary Fig. 3a). We identified 561 cells in the myoD+ cluster. Of these, 418 cells (74.5%) expressed myoD (count >0). Of the 418 myoD+ cells, 394 cells express rhGPCR (count > 0). This represents ~70% of the total number of cells in the myoD+ cluster.

6) In fact, I think that all dot plot figures from each of the clusters (from Fig. 2A through Fig. 6A) should be accompanied by separate t-SNE gray-colored plots, where the cells that express each of the two or three marker genes (selected as markers from RNA-Seq data) are highlighted in red in a separate t-SNE plot for each marker gene. In this way, one can see that the marker genes were indeed detected as simultaneously expressed in the same population of cells in the cluster.

To supplement the main figures, we've now included the requested red/grey feature plots as Supplementary Fig. 3.

7) On line 129: "Both genes showed a scattered expression pattern throughout the schistosomula (Figure 2E), [...]". This sentence refers to both rhodopsin GPCR and myoD being expressed throughout the schistosomula; however, myoD FISH data was NOT shown anywhere in schistosomula, only in adults (Fig 2F). Please correct the text in the Results and the scheme in Fig 2H.

We thank the reviewer for pointing this out. We have now included myoD FISH validation in Supplementary Fig. 4c and updated the manuscript to reflect this change.

8) Lines 159-160: “We found that cells expressing *meg17* also expressed the known oesophageal marker *meg4* (Ref. 62) [...]”. Reference 62, namely Wilson et al. (2015) is cited for MEG4 known oesophageal marker. Please note that although Wilson et al. (2015) show that MEG4 gene is among the 27 MEGs that were detected by RNA-seq to be enriched in a homogenate preparation of adult male heads compared with a homogenate of isolated adult tails, that paper does not show MEG4 as an oesophageal marker. The paper by DeMarco et al., 2010, *Genome Research* 20: 1112-1121 shows by whole mount in situ hybridization that MEG4 is expressed in the primordial oesophagus of day 3 schistosomula. This work should be cited instead.

Also, in the scheme of Fig 3H the paper of Dillon et al. 2007 is cited for MEG4 in the esophageal gland, and it should be replaced by DeMarco et al., 2010 (no MEGs were probed in the Dillon et al. 2007 paper, only protein-coding genes with known functions).

We thank the reviewer for pointing this out. The correct literature is now cited in the manuscript (Fig. 3h).

9) On lines 162-163 the authors stated: “Distinguishing the second tegumental cluster was challenging due a lack of Tegument 2-specific markers (Figure 3A).” I noticed in Figure 3A that *mboat* gene (*Smp_169040*) was enriched in Tegument 2 as opposed to Tegument 1 and it could have been used as a distinguishing marker. *Mboat* belongs to the family of Membrane Bound O-Acyltransferase proteins, an enzyme that acylates membrane lysophospholipids. This enzyme could eventually be related to the acylation of specific phosphatidylinositol membrane phospholipids that may be required for clathrin-related endocytosis. The next paragraph (lines 171-179) discusses components of the clathrin pathway identified in Tegument 2; this enzyme could be cited as possibly related to the clathrin pathway.

We’re grateful to the reviewer for drawing greater attention to *mboat*. Although it is not highly expressed, a follow-up FISH experiment has confirmed that it is a reasonably good marker for the rather challenging tegument 2 cluster. We have reworked the text describing tegument 2 and include FISH data for *mboat* in somules and adults including double FISH with *meg4* and a marker for Tegument 1. To make room in the manuscript, we no longer describe *nmda*, which was highly expressed but was a less clear-cut example.

“Distinguishing the second tegumental cluster was challenging due to a paucity of Tegument 2-specific markers (Fig. 3a and Supplementary Fig. 3b). Nonetheless, we selected two genes for further investigation: *ccdc74* (*Smp_030010*) and *mboat* (*Smp_169040*). We confirmed tegumental assignment of *ccdc74* with dextran+⁶ labelling (Supplementary Fig. 5b) and a double FISH experiment showed colocalization of *ccdc74* with Tegument 1 marker *Smp_022450* (Fig. 3f). However, distinct regions of expression for these two markers were also evident (Fig. 3f). Expression of *mboat* was lower but mostly distinct from Tegument 1 cells (Fig. 3g). Nonetheless some level of co-localisation with tegument 1 markers was still observed (Fig. 3g and Supplementary Fig. 5c). This is consistent with the expression profile for this gene that shows low-level expression in the Tegument 1 population (Fig. 3a and Supplementary Fig. 3b). *mboat* also co-localised with the oesophageal marker *meg4* (Fig. 3h) but with lower expression than *meg17* (Fig. 3e).”

As the reviewer pointed out, it is tempting to speculate that mboat is involved in clathrin related endocytosis and this point is now picked up in the Discussion.

“Further, the most discriminatory marker that we found for the Tegument 2 cluster likely acylates membrane lysophospholipids^{78,79}. It is tempting to speculate that mboat acylates the specific phosphatidylinositol membrane phospholipids required for clathrin-related endocytosis.”

10) In Figure 4F Caffrey et al. 2004 is cited for cathepsin B' as a known marker for parenchyma. Please note that the paper is not in the references list. Caffrey et al. Trends Parasitol., 20 (2004), pp. 241-248 must be cited.

We have updated the reference list accordingly.

11) I have already mentioned before, and I will repeat myself here. How many cells were detected in each of the three neural-associated clusters (line 238)? 7b2+ cells appear to be more abundant than the other two types of neural cells, according to the ISH images. Does the number of cells detected in the 7b2+ cluster reflect the apparent higher number of 7b2 positive cells in schistosomula? Please comment in the text.

We have updated the cell numbers in the manuscript as follows:

“We identified three distinct populations that expressed neural-associated genes (Fig. 6a). One population (218 cells) was characterised by the expression of genes encoding neuroendocrine protein 7B2 (7b2, Smp_073270)...A second population (20 cells) expressed the uncharacterised gene Smp_203580...we identified a population (73 cells) of cells that expressed gnai (Smp_246100), a gene encoding a G-protein G(i) alpha protein.”

The referee raises the issue of cellular abundance in the sequencing data not necessarily reflecting the true abundance in the tissue. This is certainly true for the neuronal cells. For the 7b2+, Sm-kk7+, and gnai+ clusters, we detected 218, 20, and 73 cells from the sequencing data, representing 10, 4 and 1% of quality-controlled cells, respectively. But using FISH, we detected approximately 12, 6 and 6 cells, representing 1, 0.6 and 0.6% of each type in the tissue. Neuronal 7b2+ and gnai+ cells are therefore enriched in our dataset, perhaps due to the cells being relatively resilient to treatment conditions compared with other cell types.

12) Lines 254-255: “[...] marker gene encoding KK7 (Smp_194830), known to be associated with the peripheral nervous system in *S. mansoni* (Ref. 55). Please note that Ref 55 is badly formatted in the references list. The author’s name is misspelled. It should read "Manuel, S. J." not "SJ, M.”

We have fixed this reference as indicated.

13) The use of random forest (RF) machine learning (line 270) to compare *S. mansoni* with *Schmidtea mediterranea* clusters obtained with single-cell RNA-seq is an interesting approach.

However, the Methods related to RF (lines 508-520) are confusing. Specifically, it is not clear if the evaluation of the RF classifier on the *Schmidtea* dataset (line 508) was performed with only the 692 orthologous genes between *S. mansoni* and *S. mediterranea* mentioned in lines 519-520. Since the RF classifier used for the *Schistosoma* dataset contained only those orthologous genes, it seems to me that the initial evaluation of the RF classifier on the *Schmidtea* dataset should also be done with only those 692 genes. Please clarify.

We agree with the reviewer that the section describing our use of Random Forests to classify genes was unclear.

Our first step was to test the performance of the random forest (RF) model on the *S. mediterranea* dataset of 21,612 cells using 692 variable genes common to both *S. mediterranea* and *S. mansoni* in our dataset. The model was formed by randomly selecting 70% of *S. mediterranea* cells from each cluster. This trained model was then evaluated using a 'test set' composed of the remaining ~30% of cells in the *S. mediterranea* dataset (Supplementary Fig. 10). The trained model was then used to classify the *S. mansoni* dataset (Fig. 7). In our original model, a "not assigned" category was not included. This had the effect of forcing cells into the psd+ and neoblast1+ categories. We have now adjusted the model so that a last column is included allowing cells to be classified as not fitting other categories (not assigned) (Fig. 7).

In our analysis, we also worked with 22 clusters as identified by Seurat. We then annotated each cluster using categories from Plass et al, 2018 (PMID:29674432). As reviewer 3 noted, the number of clusters annotated was lower than the numbers reported by Plass et al, 2018. This resulted in several cluster labels per population in the previous version of Fig. 7.

We originally trained our model using all the labels in the *Planaria* data set described by Plass et al, 2018. However, this model performed poorly on cross-validation tests, where many populations of cells that are highly similar to each other could not be distinguished by the model. This is particularly clear for the various categories of neoblast (Additional information 3A). Notwithstanding this shortcoming, when we applied this model using the original *Planaria* label to our *Schistosoma* data, we were able to obtain our main finding that the population Sm-kk7 strongly resembles *planaria* otf1+ cells (Additional information 3B).

Additional Information 3. Quality control metrics for the RF analysis using all labels identified by Plass, *et al.*, 2018. **(A)** The performance of a random forest (RF), trained on part of the *S. mediterranea* dataset (with clustering labels from Plass *et al.*, 2018), and used to classify a test set of *S. mediterranea* cells. The training set was formed by choosing 70% of the cells from the entire dataset, with proportional representation from each cluster. The trained RF model was then used to classify each cell in the remaining 30% of the data into learned cluster labels. Each dot represents the percentage of cells from known clusters in the test set that could be successfully categorised using the trained RF model. Cells in highly similar groups such as neoblasts, secretory and progenitors cannot be predicted. **(B)** The performance of the RF model, trained on *S. mediterranea*, used to assign cells within each schistosomulum cluster into categories. The colours and size of the circles represent the proportion of cells from each cluster (y-axis) that matches each *S. mediterranea* category label (x-axis). Only categories that received a maximum vote by a margin of > 16% of trees during the prediction are included. Cells that could not be assigned to any class are shown in the last column as 'not assigned'.

Based on the poor performance of many of the original labels on cross-validation (Additional Information 3A), we re-annotated the Planaria data as described in our methods. We assigned identities to cells based on the annotation from Plass, et. al, 2018³¹ with the following changes for a total of 30 populations:

- Neoblasts 1-13 grouped together as a single category labelled as “neoblasts”
- Activated early epidermal progenitors and epidermal neoblasts combined into “activated epidermal neoblast/progenitors”.
- Chat neurons 1-2 combined into “Chat neurons”.
- Early/Late epidermal progenitors and epidermal DVb neurons combined into “epidermal DVb neoblast/progenitors”.
- Secretory 1-4 combined into “secretory”

By pooling highly similar groups together the performance of the model is greatly improved. We have included the cross-validation of the improved model as a new Figure in the Results:

“To compare clusters, we used a random forest (RF) model trained on *S. mediterranea* (Supplementary Fig. 10) to map gene expression signatures between both datasets⁶⁷.”

We have also replaced Fig. 7 with a new one, showing the performance of the improved new model against the *S. mansoni* data and have edited the manuscript to describe the use of the “not assigned” category.

The results described in the manuscript have been simplified. We have the text from:

“We found that *Sm-kk7+* cells in schistosomula mapped to the neuronal population annotated as *otoflerin 1 (otf1+)* cells described by Plass et. al³¹. In addition, *7b2/pc2+* cells in *S. mansoni* mapped to *spp11+* and Chat neurons as well as neural progenitors in *S. mediterranea* (Fig. 7). In addition, tegument clusters in *S. mansoni* mapped to early and late epidermal progenitors in *S. mediterranea*. The rest of the clusters in *S. mansoni* were labelled as *psd+* cells (of unknown function in *S. mediterranea*) and neoblasts.”

to:

“We found the strongest similarity between *Sm-kk7+* cells in schistosomula and the neuronal population annotated as *otoflerin 1 (otf1+)* cells described by Plass et. al³¹. We also found other weaker signatures. For example, *7b2/pc2+* cells in *S. mansoni* mapped to *spp11+* neurons whilst tegument clusters mapped to epidermal neoblasts/progenitors in *S. mediterranea*. In addition, we observed that some cells on the schistosomula muscle populations mapped to *S. mediterranea* muscle progenitors”

The three sections of Methods relating to the RF approach have been rearranged to improve readability and some additional detail has been added on how the method was implemented.

14) On lines 330-331 the authors discuss about the challenges that they faced, pointing that in spite of the challenges they successfully characterised several previously unknown marker genes and populations. I would add another difficulty possibly related to the fact that an incomplete reference transcriptome was used. It has already been shown that *S. mansoni* expresses thousands of long non-coding RNAs (lncRNAs) (Vasconcelos et al. 2017, Sci Rep. 7: 10508; Liao et al. 2018, Exp. Parasitol. 191: 82–87) and that a few hundred lncRNAs are specifically expressed in schistosomula (Maciel et al. 2019, Front. Genet. 10: 823). Also, some lncRNAs are gene markers of different cell populations of single-cell juvenile and mother sporocysts stem cells (Maciel et al. 2019, Front. Genet. 10: 823). It is possible that lncRNAs are also gene markers for the different tissues that were isolated and characterized here. A future re-analysis of the single-cell RNA-Seq data obtained here that includes the lncRNAs might help to uncover some cell types that were not identified by only looking at the protein-coding genes. I think that this aspect is missing from the Discussion.

It is true that lncRNA annotations are largely missing from the reference dataset. Our analysis therefore focussed on the role of protein-coding genes. We completely agree with the reviewer that, as the reference annotation improves, it will be possible to re-analyse the data from the present study to further distinguish the transcriptomes of different cells. Following the reviewer's suggestion, we have now included a couple of sentences and a new reference at the end of the closing paragraph in the Discussion as follows:

“We necessarily focussed on gene expression changes amongst protein coding genes because these are now well annotated on the reference genome and can be quantified. This is an essential first step in unravelling the developmental biology of this important parasite. Long non-coding RNAs (lncRNA) in *S. mansoni* have also been identified from transcriptomic datasets⁸⁶. As the definitions of these additional RNA genes are included into the reference genome annotation, future reanalyses of our single-cell data may add further dimensions to the transcriptional dynamics and identify further markers of developing cells, tissues or indeed new cell types.”

Reviewer #2 (Remarks to the Author):

Authors used scRNAseq to characterise 2-day schistosomula transformed mechanically by using 10X Chromium technology, followed by validation of the cell clusters by RNA in situ hybridization (ISH) in schistosomula and adult worms. This study resulted in 11 discrete cell populations identified and novel marker genes validated for muscles, nervous system, tegument, parenchymal/gut primordia and stem cells. This study enhance a better understanding of cell types and tissue differentiation in schistosomula and is essential for unravelling the developmental biology of this important parasite.

We thank the referee for such positive comments.

However, I think it would have been much stronger if the authors would have used 14-28 day old juvenile worms from mice and not simply cultured schistosomula (2-day old, which have different biochemical parameters from the in vivo grown worms). The authors could have also validated their findings, running a side by side experiment using ex vivo worms as well as schistosomula to show behave similarly.

In this study, we deliberately chose to target the earliest stage of schistosomulum development. The referee suggested we use 14-28 juvenile worms from mice but this would be a radically different scope and it is not clear how some of the major experimental challenges could be overcome. While we agree with the referee that this would be extremely interesting, worms from days 14-28 cover a range of developmental stages and can only be obtained from mice as asynchronous mixtures. Rather than the preparing cells from asynchronous mixtures, we have characterised the organism at the start of its complex developmental programme. A previous study established that mechanically and skin-transformed schistosomula have similar transcriptional profiles (Protasio, et al., 2013, PMID: 23516644), so at this very early part of the intra-mammalian life cycle we can be confident that steady-state RNA levels in schistosomula are very similar in vitro and in vivo. As the life cycle progresses these differences become more pronounced.

Previous bulk RNA-seq studies have demonstrated that ~11,000 genes presented in *S. mansoni* schistosomula (PLoS Negl Trop Dis 7(3): e2091), however, in this study, by using single cell sequencing technology 900 genes per cell (from 2,144 cells) were identified and some typical cells were not successfully isolated from schistosomula cells, my concerns are:

1) Is Liberase DL suitable for isolation schistosome cells? Results showed total 33 genes were identified from schistosomula tegument cells, excluding a number of typical tegument genes (such as TSP-2, triose-phosphate dehydrogenase) which have been demonstrated the surface location in the parasites. That indicates that Liberase DL may cause damage of tegument cells or other parasite cells.

In the original version of the manuscript we provided a comprehensive list of markers using two different statistical tests. All reviewers found this to be confusing and we agree. We now only include one list of markers (Supplementary Table 2).

The list of 33 tegument markers possibly refers to previous TableS2-Top marker genes. In this list, we highlighted some markers that came as the most discriminatory in the Seurat ROC test. This list does not include all the marker genes. Nonetheless, we do find *tsp-2* (Smp_335630; previous ID Smp_181530) and triose-phosphate dehydrogenase (Smp_056970) in that list. In particular, *tsp-2* is highly enriched in the tegument but, as the figure below shows, its expression is not limited to the tegument clusters (Additional Information 4). The situation is quite similar for triose-phosphate dehydrogenase. It is clearly highly expressed in the tegument 1 cluster but is also expressed across other clusters (Additional Information 5).

Additional Information 4. t-SNE plot showing the expression of *tsp-2* (Smp_335630) in the dataset

Additional Information 5. t-SNE plot showing the expression of *triose-phosphate dehydrogenase* (Smp_056970) in the dataset

2) The stress of schistosomula cells induced by the processing of generating alive single-cell suspension including enzymatic digestion of schistosomula and con-staining with FDA and PI and cell sorting, may result in the changes of gene profiles of cells.

We completely agree with the reviewer. Published studies (van den Brink et al. 2017, PMID:28960196; Adam M et al. 2017, PMID:28851704) have established that enzymatic digestion of tissues can induce artefacts that impact scRNA data interpretations. Based on these reports, we limited the duration of the tissue digestion step to 30 min to reduce stress. We have also examined the expression of mitochondrial and stress-related genes during the bioinformatic analysis – standard steps that are recommended for single-cell sequencing analysis (Ilicic et al. 2016, PMID:26887813). In addition, our stringent QC based on SC3 metrics, excluded almost 1000 cells from the final analysis. See our response to reviewer 1 regarding clustering analysis (Comment 3) and Additional Information 2A-B.

A thorough FISH-based strategy to validate our bioinformatics findings was also a key part of our study, not only in the target developmental stage (2 days old schistosomula), but also using adult worms. Despite these mitigation steps, the presence of stress-related signals due to the disassociation and co-staining protocol cannot be completely ruled out. We have therefore included the following text in the Discussion:

“Signals from damaged or dying cells caused by laboratory procedures are challenging to eliminate^{81,82} but we have followed a stringent FACS-based selection protocol to enrich for live cells. Like others, we have looked at the expression of mitochondrial genes and stress-related genes, as recommended for single-cell sequencing analysis⁸³ and our bioinformatic findings are supported by FISH-based validation in both schistosomula and adult worms ”

3) In this study, approximately 7000 cells per reaction were supposed to be loaded for single cell sequencing, actually, only about 1200 cells per experiment were captured. That may lead to lose information from uncaptured cells, which brings difficult to obtain RNA information from rare cell populations or from rare genes with low gene copy number. Can you explain why there is only limited cells were captured for the analysis? Do authors have any ideas to improve the protocol? It is necessary to optimize current method due to in-depth transcriptome analysis requires the profiling of a large number of cells.

We agree with the reviewer that the number of cells captured differs from the number expected. There may be a number of reasons for this. We used a Chromium 10x 3' V2 table that provides the optimal cell concentration to obtain a specific number of cells. The technical note (CG000108) released by 10X Genomics assesses the impact of different cell concentrations on the target recovery rates. These calculations are based on single-cell suspensions from HEK293T cells. These cells are probably easier to count and represent a best-case scenario.

In addition, it may be that Chromium chemistry for version 2 could negatively impact target recovery rates. For example, newer kits with more updated chemistry (Chromium 10x 3' V3) have yielded better target recovery on a parallel project in our lab (on mouse epithelial cells).

Cell preparation was the most challenging step of our single-cell protocol. In order to obtain single cells, the tissue had to be dissociated and live single cells selected by FACS sorting. As a result, we expected a number of cells to die during the transition from the FACS to the 10X machine and during experimental handling. We also hypothesise that the disparity in the number of target cells is a result of overestimating the number of cells per ul.

We took steps to minimise the loss of cells by priming eppendorfs and pipette tips with media and FBS prior pipetting. We also kept the samples on ice at all times.

Currently, we are using a mild enzyme (Liberase DL) to dissociate but we are working to improve that protocol to include psychrophilic proteases. In future, we will be working to benchmark reagents that could help us fix the tissue prior dissociation. However, given that fixing chemicals such as methanol can potentially induce artefacts in the transcriptome of samples (10x representative verbal communication). Future experiments would have to take fixation artefacts into account.

4) As known, read counts observed are affected by a combination of different factors, including biological variables and technical noise. Critically, the small amount of starting material used in scRNA-seq may amplify the effects of technical noise.

However, given the high cost of scRNA-seq and the lack of standards in regard to analyzing it, different experiments (other than ISH) to bolster the main conclusions are needed.

We are not really sure what the reviewer is suggesting here. Our results include biological replication and for validation, ISH is really the gold standard. We have validated 25 markers (including additional ones in response to reviewer comments) but given the challenging nature of schistosomes, with its lack of genetic engineering tools and absence of cell lines, further validation would be a major undertaking.

5) Can you give details about library preparation for the scRNA-seq?

The library preparation was carried out on-site by the Library Preparation Team at the Wellcome Sanger Institute. The manufacturer's protocol published in their "Single Cell 3' Reagent Kits v2 User Guide" (available from the 10X Genomics website (<https://support.10xgenomics.com/single-cell-gene-expression/index/doc/user-guide-chromium-single-cell-3-reagent-kits-user-guide-v2-chemistry>)) was followed to create gel in emulsion beads (GEMs) containing single cells, hydrogel beads and reagents for reverse transcription. After barcoded cDNA synthesis, the GEMs are broken and the cDNAs pooled. The library construction (following GEM breakage) is prepared using 10X reagents following the protocol mentioned above. We have included details of the library preparation in the manuscript:

"The 10X Genomics protocol ("Single Cell 3' Reagent Kits v2 User Guide" available from <https://support.10xgenomics.com/single-cell-gene-expression/index/doc/user-guide-chromium-single-cell-3-reagent-kits-user-guide-v2-chemistry>) was followed to create gel in emulsion beads (GEMs) containing single cells, hydrogel beads and reagents for reverse transcription, . perform barcoded cDNA synthesis, and produce sequencing libraries from pooled cDNAs The library construction (following GEM

breakage) was prepared using 10X reagents following the “Single Cell 3’ Reagent Kits v2 User Guide”.”

Reviewer #3 (Remarks to the Author):

In this work, Soria et al. present the first complete atlas of the schistosomulum, the first intra-mammalian development stage of the parasite *Schistosoma mansoni*. The authors use an in vitro system to obtain schistosomula and generate an atlas containing more than 2000 cells using 10X genomics sequencing technology. In this atlas, they identify 12 clusters, 11 of them representing transcriptionally different cell types, which they validate in schistosomula and adults using ISHs. Additionally, they compare the clusters obtained to those from *S. mediterranea*, the closest free living relative from *S. mansoni*, in order to identify similarities in the cell repertoire of *S. mansoni* schistosomulum. The work is interesting and sound, although there are some concerns that prevent me from recommending the publication of this article in its current format.

Major Comments:

- The authors mention in the text that they use SC3, Seurat and UMAP to cluster the data. However, in the methods section they just describe independent filtering and clustering analyses performed with Scater package, SC3 and Seurat, but they do not describe how these methods have been integrated, nor how they used UMAP for Clustering. Considering that clustering is one of the key steps in single-cell transcriptomics analyses, I would ask the authors to provide additional information about how the clustering has been performed. For instance, they should provide information about how they selected the genes used for clustering, the selection of PCAs (if applicable) for each method, and the integration of the methods to generate the final set of clusters.

We agree with the referee that the information on the analysis was not clear and we have changed the manuscript to make the information more accessible. Reviewer 1 raised the same issue. See our response to reviewer 1 regarding clustering analysis (Comment 3) and Additional Information 2A-B.

- The authors describe an ambiguous cluster that represents around ~10% cells obtained. They claim that it could not be “experimentally defined”. Yet, the authors do not describe which are the methods that have been used to characterize this cell population nor include it in any of the future analyses comparing the different clusters included in the main manuscript nor in the supplementary materials. According to the markers identified by the authors (Tables S2-S6), the ambiguous cluster expressed specific marker genes that could be used to validate this cell population using ISH. Thus, the authors should include the results of the experimental validation of this cell cluster as well as in the rest of the plots in the manuscript (Fig1-7) in order to understand the relation of this cell population to all the other populations identified. This will be necessary to understand if this is a new cell population or rather low quality cells, doublets or other artifacts.

There are several reasons why we named this cluster as ambiguous. Most compelling was the lack of specific markers. From the top markers identified by Seurat, only one: Smp_132210 shows a degree of specificity (Additional Information 6A). The rest of markers are enriched but not specific to this cluster (Additional Information 6A). For any given gene in the unknown cluster, that gene is also expressed to some degree in at least 70% of the cells in the rest of the dataset. This has implications for validation with FISH. Unlike other clusters, the genes expressed in this cluster did not belong to an obvious grouping based on their annotations.

FISH validation was attempted for hypothetical (Smp_176110), *serine/threonine phosphatase* (Smp_004810), *ribonucleoprotein k* (Smp_247020), *glutamine synthetase* (Smp_091460) and *f-box* (Smp_132210) but expression could only be confirmed for the gene *f-box* (Smp_132210) (Additional Information 6B-F). In adults, *f-box* (Smp_132210) was found to co-localise with *actin-2* muscle. No co-localisation was found with markers for stem cells, parenchyma or tegument (Additional Information 6B). The co-localisation with *actin-2*, stem cells, parenchyma and tegument did not work in schistosomula. As a result, we could only validate the expression in somules as single FISH (Additional Information 6B). Based on GO enrichment analysis (Supplementary Fig. 2) we only found a signal for peroxidase activity. In summary, the lines of evidence from FISH, annotation and GO analysis were weak and unable to support a clear and coherent hypothesis of the function for this cell population. ‘Ambiguous’ therefore seems to be a fair description. We chose not to include these data in the main or supplementary figures but these could be included if it is felt that they improve the manuscript. Nonetheless, we are including the list of marker genes (Supplementary Table 2) for all of the populations, including this ‘Ambiguous’ cluster for future reference.

Additional Information 6. Ambiguity of the unidentified cluster (A) t-SNE plots showing the expression of top marker genes identified by Seurat for the ambiguous cluster highlighting the lack of specificity. (B) ISH validation for *fbx* (Smp_132210) in schistosomula (left) and in adults (right panel). (C) Double FISH for *fbx* (Smp_132210) with *histone 2b* marker for stem cells in different parts of the adult tissue. From left to right: soma anterior, testes, ovary and vitellaria. (D) Double FISH for *fbx* (Smp_132210) with *tsp2* tegumental marker in the soma of the parasite. (E) Double FISH for *fbx* (Smp_132210) with *serpin*, a paranchymal marker. (F) Double FISH for *fbx* (Smp_132210) with *actin-2* muscle marker.

The authors define a positional muscle cluster based on the expression of the uncharacterized gene *Smp_161510* and *wnt* (*Smp_167140*). However, according to the data provided, there is no coexpression of these markers in the schistosomulum (Figure 2B, C), nor in adult worms (Figure 2D, Supplementary Figure 2). Thus, it seems that this *wnt*+ population may be a distinct population of cells that the authors have not been able to capture in their clustering. The authors should provide tSNE plots showing the expression of these two genes in the single-cell dataset as well as co-stainings showing the co-occurrence of the two markers.

We thank the reviewer for highlighting these points. Fig. 2a shows that *wnt-2* (*Smp_167140*) is expressed in a subset of cells (62 in total) out of 365 cells in the Positional muscle cluster. We were not able to confirm co-expression of *Smp_161510* and *wnt-2*. However, to definitively demonstrate that *Smp_161510* and *wnt-2* (*Smp_167140*) are expressed in muscle cells, we used a pan-muscle marker *troponin* (*Smp_018250*) (Supplementary Fig. 3a) to perform double FISH with these markers. The majority of *Smp_161510*+ or *wnt-2*+ cells co-localise with *troponin* in adults (shown in Fig. 2c and 2f), supporting that these cells are part of muscle cells. We therefore confirm that *Smp_161510* and *wnt-2* (*Smp_167140*) are expressed in muscle but we cannot rule out the possibility that the *wnt-2*+ cells are a distinct cluster that only differs in the expression of that marker.

- The authors should explain how they define the set of marker genes shown in Figure 3A as well as in the rest of panel A for all main Figures. According to table S4, Tegument 2 cell type has significant marker genes not present in tegument 1 cell such as *Smp_074570* and *Smp_169460*. Besides, the authors also show tegument 2 specific genes in Supplementary Figure 3H & F. The authors should provide validation of these markers or other exclusive markers from tegument 2 in order to characterize the differences across the two cell populations.

The text describing the Tegument 2 cluster has been rewritten to improve clarity and to include more FISH validation for markers specific to Tegument 2. In particular, we now include *mboat* (*Smp_169460*), *epsin-4* (*Smp_140330*), and *dynammin* (*Smp_129050*) as suggested by the reviewer as validated markers (Fig. 3g-h; Supplementary Fig. 5c,5f-h; Supplementary Fig. 6c-e).

A major factor for choosing the original list of markers was the AUC values given by the roc test from Seurat. The original validated marker list included the top genes with AUC values > 0.8. The only exception to this was *wnt-2* (AUC=0.57), which we discovered by filtering for genes that were expressed in the specified cluster with almost 0 expression anywhere else in the dataset. As with *wnt-2*, we took into consideration the expression of the marker gene in the cluster that it was supposed to discriminate, versus every other cluster in the dataset. For populations such as muscle for example, we looked at how good that particular gene was at discriminating between the muscle populations and also between all the muscle versus the rest of the populations. In parallel, we also plotted the expression of the marker genes to visually inspect how good that specific marker gene was at discriminating between clusters (Supplementary Fig. 3).

- The authors should make a unified set of marker genes instead of reporting three independent sets of marker genes (Tables S3-S5).

We thank the reviewer for the comment and have now provided just one list of marker genes. Please see earlier response to reviewer 1 and 2

- The authors used a Random Forest classifier to identify similarities across schistosomula and *S. mediterranea*. The approach is interesting but the authors should explain the performance of the method. For instance, they report that they evaluate the classifier using the Schmidtea dataset but they do not report the performance of the method, i.e. the accuracy of the classification. Additionally, the authors should explain how they annotated the clusters from *S. mediterranea* given that the original publication contained a different number of clusters. The authors show in Fig. 7 that most of their identified cell types correspond to “psd+ cells: neoblast1” identified in *S. mediterranea*. Psd+ cells is a rather small cluster in the original publication whereas neoblast1 is the main stem cell group identified. The authors need to reinterpret the correspondence of clusters as it seems that most of them have stem cell related genes rather than psd+ cells genes.

We thank the reviewer for pointing this out. Previous reviewers have also commented on these points (Please see reviewer 1, comment 13 response).

- In the discussion, the authors acknowledge the possibility of missing known cell populations because they cannot identify a specific cluster containing them, such as the case of the protonephridia cells. To support their claims, the authors should provide some plots showing the expression of known protonephridia marker genes in the tSNE plot.

Following the reviewer’s comments, we have now included tSNE plots highlighting known protonephridial markers (Supplementary Fig. 11). The bone morphogenic protein (BMP) homologue (Smp_343950) in *S. mansoni* previously described as protonephridial tubules marker (Freitas, *et al.*, 2009; PMID: 18765241), is expressed in the muscle and nervous system (Supplementary Fig. 11). The same pattern of expression in the muscle and nervous system is observed in adults (Wendt, *et al.*, preprint: <https://doi.org/10.1101/2020.02.03.932004>). This has been shown in *S. mediterranea* where relatively rare cell types are sometimes embedded in larger neuronal clusters (Plass, M. *et al.*, 2018, PMID:29674432; Fincher, C. T. *et al.*, 2018, PMID:29674431). In addition, another gene Smp_079770, an ER 60 homologue previously associated with the lining of the protonephridia (Finken-Eigen and Kunz, 1997; PMID: 12204221), was found to be widely expressed in all clusters (Supplementary Fig. 11). Two flame cell markers, Smp_335600 and Smp_035040 (Wendt *et al.*, 2020; doi:10.1101/2020.02.03.932004) were only expressed in a very few cells in the dataset (Supplementary Fig. 11). Accordingly, we have updated the Discussion to include this item as follows:

“Notwithstanding these measures, some known cells were not detected, possibly due to their rarity in the schistosomula or fragility during tissue dissociation. The absence of eight protonephridia cells, known to be present in schistosomula^{11,84}, is a notable example. In our data, the *S. mansoni* bone morphogenic protein (BMP) homologue (Smp_343950)⁸⁵, a previously described protonephridial marker, was found in the muscle and nervous system (Supplementary Fig. 11).”

Minor Comments:

- The authors report the number of reads per cell. This number is not relevant given that it could reflect only sequencing of PCR duplicates. The authors should report instead the number of UMIs per cell.

We agree with the reviewer and we have now included UMIs per cell in Supplementary Table 1

- Why do the authors use characterized genes in human and mouse to define cell populations and not those from closer relative species?

For the Tegument and Stem cells, we used genes that had been characterised for *Schistosoma mansoni*. For muscle and some neuronal populations (except *Sm-kk7+* cells) where we did not have many validated gene markers in *mansoni*, we decided to use marker genes with a high level of functional characterisation in the literature and these happened to be human. We did however compare all our clusters to *Schmidtea* using a Random Forest analysis.

- Table S3 lacks the association of ~3000 genes to identified clusters

Thank you for pointing this out. We have now included only one list of marker genes with the corresponding cluster labels

- The authors should provide additional details about how they define top markers (included in table S2) as well as which are the parameters used to define markers with Seurat.

As all reviewers have pointed out, the different marker genes lists provided were not clear. For completeness, we included lists from SC3 and for two Seurat tests (ROC and Wilcox). However, we agree with the reviewer that this was confusing, and have simplified the manuscript by only retaining the list of marker genes calculated using the ROC test. The latter provides a confidence (AUC) value on how discriminatory each marker should be for marking a particular cluster. This was key information that we used when manually selecting marker genes for validation. The specific parameters we used for the FindAllMarkers function were: `test.use="roc"`, `only.pos = TRUE`, `return.thresh = 0`.

We have now updated the methods to include this extra information. The text was changed

from:

“We also used the Seurat package to identify marker genes for each population using the function FindAllMarkers, using the likelihood ratio as specified in the Seurat best practices (<https://satijalab.org/seurat/>).”

to:

“We used the Seurat package to identify marker genes for each population using the function FindAllMarkers and `test.use="roc"`, `only.pos = TRUE`, `return.thresh = 0`, as specified in the Seurat best practices (<https://satijalab.org/seurat/>). We used the

‘area under the roc curve’ (AUC) > 0.8 value and spatial information of those genes to determine the identity of a specific population.”

- Figure 2A in line 134 should be Figure 2G.

The correct figure (now Fig. 2i) is now cited.

- I cannot find the ISH pictures of the validation of fimbrin in Fig 2 or Supp. Fig 2.

The correct figure (now Supplementary Fig. 6c) is now cited.

- The authors should define if the ISH images report dorsal or ventral views.

We have indicated dorsal or ventral views wherever necessary in figure legends, but not for all ISH images. Images that show maximum intensity projection of schistosomula do not look different whether they are shown from dorsal or ventral views, since the dorso-ventral information is lost upon projection to the two-dimensional space.

- The authors should provide tSNE plots showing the expression of all the validated marker genes.

We have provided the tSNE plots of all validated genes. Please see Supplementary Fig. 3

- The zoomed area in Figure 3G does not correspond to the region highlighted in the general FISH picture on the left.

We removed *nmda* FISH data from the manuscript, since it was not the best gene to differentiate between Tegument 1 and 2. Instead, we added marker genes that are more enriched in Tegument 2, including *mboat*, *epsin-4*, and *dynammin* (Fig. 3g and 3h, Supplementary Fig. 5c, 5f-h, and Supplementary Fig. 6c-e).

- I do not see coexpression of *Smp_022450* and *annexin B2* in schistosomulum heads.

We confirmed that the two genes show co-localization in the head, but *annexin B2* at a lower level. We have clarified this by adding a zoomed single confocal section image of the head region to clearly show the co-expression (Fig. 3c).

- The ISH showing the expression of *Smp_022450* in Figure 3B is very different from the staining obtained in Figure 3D, 3F and 3G, in particular in relation to the expression of the gene in the body. The authors should explain this discrepancy.

We thank the reviewer for allowing us to clarify this point. The expression of *Smp_022450* is the strongest in the head and the upper body (neck) region, while it is lower in the mid-body area. Consistently, *Smp_022450* shows expression in the head of the somules (Fig. 3c, 3d, and 3f) and, in the body, it shows a scattered expression, albeit not exactly in the same spots across each schistosomulum shown. Given this is a marker of tegumental cells, presumably distributed across the whole body, the scattered distribution in the body makes sense. The strong and consistent expression in the head is remarkable (particularly pronounced in Fig. 3d).

One technical challenge we encountered during the image acquisition and processing was the stark difference in signal between the head and mid-body regions. As we avoid saturating out the strongest signal, weaker signals in the mid-body can appear low or absent, which might be one of the reasons for the variability between the worms.

- In relation to the analysis performed by the authors in Suppl. Figure 3, does Tegument 1 also have specific functions and thus tegument 1 and 2 represent two functionally distinct cell types of the tegument? The authors should clarify this point.

We find evidence that six genes specific/enriched in Tegument 2 are involved in clathrin-related endocytosis. As reviewer 1 rightly pointed out, the Tegument 2 marker *mboat* (Smp_169040) –now validated by FISH –likely acylates membrane lysophospholipids. For this reason, we speculate on a possible functional difference between these two clusters in the discussion. However, more functional validation will be necessary in the future to test this hypothesis.

“By analysing inferred interactions between Tegument 2 genes, a group was identified that included homologues of known vesicular transport proteins: epsins and a phosphatidylinositol-binding clathrin assembly protein. Further, the most discriminatory marker that we found for the Tegument 2 cluster likely acylates membrane lysophospholipids^{78,79}. It is tempting to speculate that *mboat* acylates the specific phosphatidylinositol membrane phospholipids required for clathrin-related endocytosis.”

- Rename the panels in Suppl Figure 4 so that they are cited in the text in the same order as in the main text.

We thank the reviewers for the suggestion. We have rearranged the panels to match the order that is presented in the text.

- Include the cell type for which Smp_022450 is a marker in Figure S4G.

We have added the ‘Tegument 1’ label next to Smp_022450 for this panel.

- Remove “that” in line 200. “genes mark schistosomula parenchyma,”.

We have removed ‘that’ as requested

- Rename panels in Figure 5 so that they appear in the same order as in the text.

We have rearranged the figure panels to reflect the order presented in the text.

- Authors claim that *cam+* cells are *h2b+* (line 222). Yet, in figure 5 they show little overlap. Please rephrase.

We agree with the reviewer and have changed the text accordingly from:

“The *cam+* cells were also positive for *h2b*”

to:

“In addition, some *cam+* cells were also positive for *h2b* in schistosomula (Fig. 5d). In adults, *cam+* cells were expressed in the adult gonads (Fig. 5e) and soma (Fig. 5f).”

- The authors should describe in detail how did they make the plots shown in Figure 5C and which are the previously described stem cell populations depicted there.

We thank the reviewer for highlighting these plots. Genes shown in previous Supplementary Fig. 5c (now Supplementary Fig. 8c) were identified by Wang *et al.*, 2018 that allowed schistosoma stem cells to be divided into three groups - *kappa* (*klf+/nanos2+*), *delta* (*nanos2+/fgfrA+*), and *phi* (*nanos2-/fgfrA+*). We wanted to find out if it would be possible to sub-cluster our stem cell population based on these three groups. We investigated this by plotting the expression of genes included in each of those stem cell classifications in our dataset. In this figure, we only visualise the stem cells to help with the visualisation of the expression of these genes using umap. However, given that we use t-SNE plots in the rest of the manuscript, we have changed the plots for consistency with the rest of the figures showing Feature plots (Supplementary Fig. 8c).

- Mark the location of the main and minor nerve cords in Figure 6G and Supplementary Fig. 6E & F.

We have clarified this in the figure panels (Fig. 6b-c, 6g and Supplementary Fig. 9e-f).

- There is a typo in Figure 6H. It says “gnia” instead of “gnai”

We thank the reviewer for identifying this typo, we have corrected it.

- In the Random Forest method description, the authors have a discrepancy in the number of cells used: from 21610 they keep 21612 cells.

We are grateful to the reviewer for spotting this discrepancy. The correct number of planarian cells is 21,612, we have fixed the text accordingly.

Additional information 1

FUGI_R_D7119553

FUGI_R_D7159524

FUGI_R_D7159525

Reviewers' Comments:

Reviewer #1:

Remarks to the Author:

The authors have submitted a considerably revised version of the manuscript and I find that all previous criticisms and suggestions have been adequately addressed. I have no further concerns.

Reviewer #2:

Remarks to the Author:

By using advanced single cell sequencing technique, this study presented distinct features of different single cell clusters represented in young schistosomula. This study is novel and timely, providing important information for better understanding of the molecular mechanism that regulates parasite development. That will be of interests to parasitologists worked in other helminths. Authors addressed my questions well.

Hong You

Reviewer #3:

Remarks to the Author:

In this revised version of the manuscript, Soria et al. present a clearer version of the text in which they address many of my concerns raised previously. However, there are still some issues that need to be further clarified.

The authors now provide many more details on the quality control and filtering of the single cell data that helps understanding the data processing and quality control. However, there are some aspects of these methods that are still quite intriguing. For instance, the authors obtain 3513 cells from the 10X Genomics Chromium Platform, from which 2,144 cells pass strictly the quality control filters. 2144 cells represent ~60% of the total cells sequenced. Given that such a high amount of cells are discarded, the authors should provide additional diagnostic plots and details to help the reader understand the importance of these filtering steps, as so far only the cutoffs are given. For instance, why cells with more than 30000 UMIs or less than 600 genes were discarded? Were these cells clear outliers?

In the same line, the authors afterwards explain how they use SC3 package to discard clusters with a stability index < 0.1 or containing less than 3 cells. This is a step that surprises me as clusters with a low stability index probably reflect an over clustering situation rather than the existence of low-quality cells within them. Yet, the authors use this approach to discard low quality cells. In this regard, the authors should better explain why they use this approach and how many cells they discard with this procedure, as the number of cells reported is the same as those reported after using the Scatter package.

In the description of how they use Seurat Package for clustering I find some information still missing. The authors should indicate the number of variable genes selected with the FindVariableGenes function and the number of principal components selected to perform the Clustering and tSNE, and how these PCs were selected (p-value/Elbow plot?).

In the response to the reviewers, the authors clarify and explain in detail the information in regard to the "ambiguous" cluster and their validation attempts. I understand that the lack of prior information or knowledge may prevent the authors to assess the identity of this cluster. Yet, these analyses show

that this cluster is likely not an artifact but an unknown cell population. Taking this into account, I think that it is important that this is clearly stated in the main text. The authors should include the ambiguous cluster in Fig 1C, Fig. 7 as well as in Supplementary Figure 3. Furthermore, they should include the ISH provided in the Additional information 6 in the Supplementary Material to support their findings.

I have not been able to find the information about how the stem/germinal cell population subclustering provided in Supplementary Figure 8. Given that the authors are not able to identify the previously described populations from Wang et al., I wonder if the authors have recomputed variable genes and PCs for this subset of cells or if they have just replotted a tSNE using these cells. If the latter is the case, the authors should perform an analysis of these cells independently and recalculate the variable genes, PCs, clusters and tSNE. This may help assessing the presence of these previously defined stem/germinal cell populations.

In relation to the Random Forest classifier, the authors now provide a more detailed description of the benchmarking and performance of the method. However, there are still some details missing. In the Additional Information 3, the authors show the performance of the RF on *S. mediterranea* data. However, in this analysis they do not include the "unassigned" category, which they show it changed significantly the assignment of *S. mansoni* cells to *S. mediterranea* clusters, nor the overall accuracy of the method. These changes have to be made in order to evaluate properly the performance of the RF.

Besides, I wonder about the striking differences between the RF classification provided in the Additional Information 3B and in Figure 7. Although the RF in Additional Information 3B includes extra clusters, I wonder how come barely none of the cells in any of the muscle clusters described in *S. mansoni* are assigned to muscle progenitors nor to the muscle body cluster and yet this correspondence is very clear in Figure 7.

In line 298, the authors state "these results suggest that despite great differences in developmental stages between larval schistosomula and the asexual adult *Schmidtea mediterranea* used for this comparison, marker genes for stem cells and neuronal populations have been conserved (Figure 7)". However, the authors use for the RF a set of 692 Variable Genes conserved between *S. mediterranea* and *S. mansoni* of unknown identity. Although amount these genes there may be some known markers of neuronal and stem cell populations, this is actually not known. Thus, the authors should provide the list of conserved variable genes used for this analysis. Additionally, to make such a statement, the authors should provide a comparison of the marker genes across cell populations in *S. mansoni* and *S. mediterranea* to show that actually some of the marker genes are conserved across species.

Minor comments:

In Supplementary Figure 4e and f it is not clear what the dashed boxes mark, as the zoomed FISH images on the right do not correspond to the left pictures. If these are single confocal images, this should be stated in the Figure caption.

In Line 193, the authors should specify that oesophageal gland cells are similar to tegument 2 cells.

The sentence "In addition, parenchymal cells did not co-express other cell type markers except for actin-2, which showed slight overlap in expression (Supplementary Figures 7F-7J)" (line 206) it is ambiguous, as it is not clear if the authors refer among parenchymal cell types or between parenchymal cell types and other cell types.

Line 370. Authors should reference Figure S11.

Reviewer #1 (Remarks to the Author):

The authors have submitted a considerably revised version of the manuscript and I find that all previous criticisms and suggestions have been adequately addressed. I have no further concerns.

Reviewer #2 (Remarks to the Author):

By using advanced single cell sequencing technique, this study presented distinct features of different single cell clusters represented in young schistosomula. This study is novel and timely, providing important information for better understanding of the molecular mechanism that regulates parasite development. That will be of interests to parasitologists worked in other helminths. Authors addressed my questions well.

Reviewer #3 (Remarks to the Author):

In this revised version of the manuscript, Soria et al. present a clearer version of the text in which they address many of my concerns raised previously. However, there are still some issues that need to be further clarified.

- 1) The authors now provide many more details on the quality control and filtering of the single cell data that helps understanding the data processing and quality control. However, there are some aspects of these methods that are still quite intriguing. For instance, the authors obtain 3513 cells from the 10X Genomics Chromium Platform, from which 2,144 cells pass strictly the quality control filters. 2144 cells represent ~60% of the total cells sequenced. Given that such a high amount of cells are discarded, the authors should provide additional diagnostic plots and details to help the reader understand the importance of these filtering steps, as so far only the cutoffs are given. For instance, why cells with more than 30000 UMIs or less than 600 genes were discarded? Were these cells clear outliers?

In the same line, the authors afterwards explain how they use SC3 package to discard clusters with a stability index < 0.1 or containing less than 3 cells. This is a step that surprises me as clusters with a low stability index probably reflect an over clustering situation rather than the existence of low-quality cells within them. Yet, the authors use this approach to discard low quality cells. In this regard, the authors should better explain why they use this approach and how many cells they discard with this procedure, as the number of cells reported is the same as those reported after using the Scatter package.

Initially, we excluded 287 cells with Scater and 1,082 with SC3 based on best-practice at the time of carrying out the work. To determine the number of clusters, we started with the SC default estimate of 26 because we had no prior knowledge of the true number to expect. We iterated through several ks values (23-26) close to the initial value and examined the cluster stabilities.

Following the reviewer's suggestions, we have now examined the output using SC3 ks values from 12-14 to reflect the number of populations that we obtained in our manuscript. With these lower starting estimates, the clustering and stability values do in fact change.

In light of this observation and the reviewer's comments, we have now simplified the methodology; we have rerun the Seurat analysis but with the SC3 filtering excluded. As before, we have excluded cells with a high number of counts (potential doublets, nUMIs > 30,000), cells with a low number of detected genes (<600) or with a high MT percentage (indicative of stressed cells, >2.5%). After these QC filters, we are now left with 3,226 cells, which we have processed and clustered using Seurat (as described in the methods).

The same clusters of cells identified previously were also found in the revised analysis, although the number of cells per cluster has increased. As a result of these changes, the number of cells per population have been updated in the manuscript and the new and old counts per cluster are summarised in Additional information 1.

Additional Information 1. Number of cells per cluster after re-analysis (new) versus previously submitted analysis (old)

clusters	Muscle			Neurons				Tegument		Meg4+	Parenchyma		Stem/ Germin al	Ambiguous	
	MyoD+	Posit.	Actin 2+	7b2/pc2	gnai+	NDF+ neurons	Smkk7+	Teg. 1	Teg. 2		1	2		1	2
Cells (new)	788	428	224	450	141	32	20	182	99	17	101	57	126	290	271
Cells (old)	561	365	179	218	73	ND	20	176	77	ND	83	72	94	226	

In addition to the changes of cell counts per cluster, three new clusters have appeared. First, there is an extra cluster of 271 cells annotated as 'Ambiguous 2' (Fig. 1 in the manuscript). For this cluster no specific markers were predicted (Supplementary Fig 12 in the manuscript) and no clear signal of GO term enrichment was evident (Supplementary Fig. 2 in the manuscript).

Second, we have found a small group of 32 cells, originally clustered with 7b2/pc2+ neurons, that after the re-analysis clustered separately. We refer to this group as NDF+ neurons because ~69% of cells in this cluster expressed a putative neurogenic differentiation factor (Smp_072470), compared with less than 0.1% in other clusters (Fig 6 in manuscript). The orthologue of this gene in *S. mediterranea* (*neuroD1*) is also associated with neuronal populations. When knocked down in *S. mediterranea* in combination with other genes, it results in a decrease in a *npp-4+* population (Cowles, M. W. *et al.*, 2013).

Unfortunately, due to the current closures of our facilities, we have been unable to validate this gene in schistosomula. Nonetheless, the gene has been validated by Wendt, *et al.*, 2020 (cluster neuron 14; <https://doi.org/10.1101/2020.02.03.932004>) in adult worms in the nervous system. Accordingly, we have now updated the manuscript to include references to this cluster as follows:

“The last population comprised 32 cells and was annotated as *ndf+* neurons. This population was characterised by the expression of a *neurogenic differentiation factor* (*ndf*; Smp_072470) and a *neuropeptide receptor* (Smp_118040) (Fig. 6a). The *neurogenic differentiation factor* (*ndf*) Smp_072470 has recently been identified as a neuronal marker in adult schistosomes⁷¹.

The orthologue of *ndf* in *S. mediterranea* (*neuroD1*) is also associated with neuronal populations⁷². Knockdown of this gene in combination with other neural specification genes in *S. mediterranea*, results in a decrease of 40% in a *npp-4+* population⁷². In addition, as well as expression in the nervous system, this gene is expressed in X1 neoblasts (stem cells) from wounded *S. mediterranea* and in cells near regenerating anterior blastemas⁷³. Although further experiments will be needed to ascertain the biological function of this population, data from adult schistosomes and *S. mediterranea* suggest this is indeed a neuronal population. Based on previous findings on *S. mediterranea*, it may be involved in the specification of other neural populations.”

The last of the three new clusters comprised 17 cells. These cells represent a putative oesophageal gland population characterised by the expression of several microexon genes (MEGs). Interestingly, one of the top marker genes for this cluster is *meg-4*. This gene showed a localised expression in the oesophageal gland of schistosomulum (Figure 3g and Supplementary Figure 6a) (DeMarco et al., 2010). We note that in the latest version of the genome (at parasite.wormbase.org), the gene is present as a near-perfect duplication (the original probed sequence is completely contained within two new gene models: Smp_307220 and Smp_307240) although our probe captures Smp_307220 with greater efficiency (Supplementary Table 4).

We have updated the manuscript to include a new section:

“Micro-exon gene expression is enriched in the oesophageal gland.

We also discovered a small population of oesophageal gland cells (17 cells) that expressed *meg4* genes (Smp_307220/Smp_307240)⁵⁸ (Fig. 3a, Supplementary Fig. 3f and Supplementary Fig. 6a-b). The oesophageal gland is an anterior accessory organ of the digestive tract⁵⁹ and is crucial for degradation of host immune cells and parasite survival⁶⁰. This group of cells also expressed other *meg* genes with high specificity such as *meg8* (Smp_172180), *meg9* (Smp_125320), *meg11* (Smp_176020), *meg15* (Smp_010550), and *meg32.1* (Smp_132100) (Fig. 3a). The function of this class of genes is enigmatic but they have the capacity to generate protein diversity based on their propensity for exon skipping^{58,61}. Given the expression of some *meg* genes around the oesophagus of adult parasites^{62,63} and the developmental relationship between the oesophagus and the tegument^{8,64}, we tested if tegumental genes co-localised with any known genes from the *meg4+* oesophageal gland population (Fig 3a.). We found that *mboat* co-localised with the oesophageal gland marker *meg4* (Fig. 3g). Similarly to *mboat*, we also observed colocalisation of *epsin4* with *meg4* in the oesophageal gland (Supplementary Fig. 6b). In adults, Tegument 2 markers such as *epsin4* (Smp_140330) and *mboat* (Smp_169040) were consistently enriched in the oesophageal gland of the adult worm (Supplementary Fig. 6c-d). In the case of *mboat*, we could observe colocalisation with *meg4* in the oesophageal gland (Supplementary Fig. 6e). Therefore, the *meg4+* oesophageal gland cells share similar molecular composition and function to the Tegument 2 cells (Fig. 3h).”

During the revision of the manuscript, we also noticed that a probe, originally cloned to target Smp_180620 *meg17+* cells, also overlapped another gene (Smp_335780) with over 50% of the sequence (Supplementary Table 4). This newly identified gene has only recently been found to be in chromosome 4 in the newest genome assembly for *S. mansoni* (manuscript in preparation). We have therefore removed the *in situ* hybridization of *meg17* in this new version (in the previous manuscript version the removed figure corresponded to Fig. 3e) since the probe may bind to both genes. We have also checked the rest of our probe sequences against this newer genome assembly and confirmed that all remaining probe sequences uniquely target their intended genes.

All the figures have now been updated based on the new findings and suggestions by the reviewer for Comment 3.

- 2) In the description of how they use Seurat Package for clustering I find some information still missing. The authors should indicate the number of variable genes selected with the FindVariableGenes function and the number of principal components selected to perform the Clustering and tSNE, and how these PCs were selected (p-value/Elbow plot?).

In our original analysis, we used 17 PCs to do the clustering analysis and 1,010 Variable genes (version 2 Seurat). PCs were selected based on a visual inspection of an Elbow Plot (Additional information 2).

Additional Information 2. Elbow plot produced by Seurat to determine the number of PCs. The optimal PC is chosen in the range where the 'elbow' has levelled off.

In our revised analysis, we have used 2,000 variable genes (version 3.1.5 Seurat finds more variable genes by default) but, given the ambiguity in determining suitable numbers of PCs from the plot, we have also used a statistical method that utilises machine learning to make a prediction of the number of PCs required for the single-cell dataset (Additional information 3). This method finds the optimal number of PCs using a molecular cross-validation method (doi: <https://doi.org/10.1101/786269>), which is applied to raw count data, to determine the optimal number of PCs to use for a given dataset. This is intended to be run as part of a Seurat workflow. The relevant R package can be found at <https://github.com/constantAmateur/MCVR/blob/master/code.R>

Based on the visual inspection of the ElbowPlot and the cross validation method, we have included the optimised number of 25 PCs. The methods in the paper have been updated to reflect these changes.

Molecular cross validation determination of optimal number of PCs

Additional Information 3. Molecular cross-validation determination of optimal number of PCs. In the **upper panel**, the x-axis represents the number of PCs used. The y-axis represents the accuracy with which a given number of PCs calculated on training data predicts the test data, as measured by molecular cross-validation, normalised so that 1 is the lowest value for each coloured curve. Different colours represent the results of this analysis using different fractions of the data, as indicated by the legends. Solid lines represent the mean cross-validation accuracy while dashed lines represent the range (min/max). In the lower panel, the y-axis represents the number of PCs used and the x-axis the fraction of data used to calculate the optimal number of PCs. Dots represent the optimal number of PCs for each data fraction (minimum in upper panel), with whiskers indicating the 95% confidence interval in the estimate of the optimal number of PCs.

3) In the response to the reviewers, the authors clarify and explain in detail the information in regard to the “ambiguous” cluster and their validation attempts. I understand that the lack of prior information or knowledge may prevent the authors to assess the identity of this cluster. Yet, these analyses show that this cluster is likely not an artifact but an unknown cell population. Taking this into account, I think that it is important that this is clearly stated in the main text. The authors should include the ambiguous cluster in Fig 1C, Fig. 7 as well as in Supplementary Figure 3. Furthermore, they should include the ISH provided in the Additional information 6 in the Supplementary Material to support their findings.

Our interpretation of the evidence is that the cluster originally labelled as ambiguous likely represents dead or dying cells for the following reasons: (1) the ISH experiments showed them to be dispersed across the whole organism; (2) no discriminatory markers were revealed by Seurat; (3) analysis of expression signatures revealed no coherent trend/enrichment. In other single cell studies, cells of this kind have been presented in numerous ways but are frequently removed or ignored. However, we do recognise that this is an area of subjectivity and have now incorporated additional aspects of the analysis into

the paper to ensure that information is not missed. We have now included the ambiguous cluster in the Random Forest Analysis as requested by the reviewer and updated all the figures of the paper as requested (Fig. 1-7, Supplementary Fig. 3, Supplementary Fig. 11-12).

As detailed in our comment above, the inclusion of more cells has resulted in an additional cluster of cells (called Ambiguous 2), where assigning to a specific cell type is not possible. Although distinct from one another, neither ambiguous cluster presents a single coherent set of GO terms that set them apart from other clusters (Supplementary Fig. 2). In addition, no specific cell markers were identified (Supplementary Fig. 12).

- 4) I have not been able to find the information about how the stem/germinal cell population subclustering provided in Supplementary Figure 8. Given that the authors are not able to identify the previously described populations from Wang et al., I wonder if the authors have recomputed variable genes and PCs for this subset of cells or if they have just replotted a tSNE using these cells. If the latter is the case, the authors should perform an analysis of these cells independently and recalculate the variable genes, PCs, clusters and tSNE. This may help assessing the presence of these previously defined stem/germinal cell populations.

Following the reviewer's comments and the changes in the number for cells, we extracted the stem cell population and analysed just these cells through the same Seurat pipeline described above. We then looked for evidence that the results recapitulated the populations validated by Wang, *et al.*, 2018. However, we cannot see a pattern that could be used to justify the splitting of these cells into distinct sub-clusters. The only common feature amongst the 3 subclusters is the expression of the main stem cell marker validated in our paper (*cam*; Smp_032950) (Additional information 4).

It is important to stress that the approach followed by Wang, *et al.*, 2018 completely differs from our own study. For example, they analysed dissociated cells from mother sporocysts transformed *in vitro*, a completely different (intramolluscan) stage to the one in our study. It is known that *in vitro*-transformed mother sporocysts contained up to 20 germline cells that give rise to daughter sporocysts within the snail; therefore, the germline:soma cell ratio in sporocysts is higher than in schistosomula. In addition, Wang *et al* employed a microfluidic RNA-seq chip (Fluidigm) system to characterise the single cell transcriptome of 35 cells. Following the scRNAseq identification of these subclusters in sporocysts, the authors then selected subcluster-specific markers to study their expression by ISH in developing intramammalian parasites, including 2-day old schistosomula.

Given that we used a different technology (10x Chromium), did not enrich for any particular population, and worked on a different parasitic stage, it is likely that we did not have enough sensitivity to capture these stem cell subclusters. We have now added the following statements in the Discussion:

“In contrast, we were unable to identify three stem cell populations (*delta*, *kappa* and *phi*) that were previously described by Wang *et al.*¹⁵. In the latter study, marker genes were identified from the single-cell transcriptomes of 35 cells obtained from sporocyst germinal centres. Some marker gene expression was subsequently confirmed in 2-day schistosomula. In our study, a particular cell population was not specifically targeted. As such, our sensitivity to identify the

reported germline subcluster markers may have been reduced, particularly given their expected low expression levels¹⁵”.

Additional Information 4. Heatmap of markers identified by Wang, et al., 2018. The stem cells were extracted from the dataset and clustered with Seurat into three clusters (using 2 PCs). For each cluster, we examined expression of markers previously used by Wang, et al., 2018 to discriminate three stem cell populations (*delta*, *kappa* and *phi*). The markers could not discriminate the expression of these putative sub-clusters; thus we could find insufficient support to cluster schistosomula stem cells into subpopulations.

- 5) In relation to the Random Forest classifier, the authors now provide a more detailed description of the benchmarking and performance of the method. However, there are still some details missing. In the Additional Information 3, the authors show the performance of the RF on *S. mediterranea* data. However, in this analysis they do not include the “unassigned” category, which they show it changed significantly the assignment of *S. mansoni* cells to *S. mediterranea* clusters, nor the overall accuracy of the method. These changes have to be made in order to evaluate properly the performance of the RF.

We thank the reviewer for the comments. In the previous figure, the assignments are spread across the entire dataset. We have now included the plot with the unassigned categories below (Additional information 5). The figure shows that most of the neoblast populations remain unassigned. In some instances, no cells are assigned to a particular population. In those cases, the class label is not shown in the x axis and remains as unassigned. We have now included this plot in Supplementary Fig. 10, as panel A, to illustrate why we decided to pool the labels of similar groups i.e. neoblasts.

Additional Information 5. Random Forest QC using all the cluster labels for *S. mediterranea* (Plass, et al., 2018).

Colour intensity and symbol size both indicate % cells in the test data that were classified. From a normalised set of categorised data from Plass, et al., 70% of the cells in each label were used for training data (Y-axis) and used to classify the remaining 30% of cells in each category from the test data (x-axis). It is important to note that the assignment is completely agnostic to the labels in the test set. For the test set, each cell was assigned a cluster label if >16% of trees converged onto a majority vote during the prediction analysis. Some cluster labels were never assigned because no cells were confidently predicted to belong to their group. Only 43 cluster labels (out of 51) that had at least one cell in its group with the specified threshold (>16% of trees converging onto a decision for assign a label to that cell) are shown on the x-axis. For some groups shown in the x-axis where cell identities were predicted, the prediction matches the 'correct' identity (24 cell labels out of 43). The rest of the groups, the cell identities did not match the expected cell identity. Cells that could not be assigned a cluster label are represented in the 'not assigned' category.

- 6) Besides, I wonder about the striking differences between the RF classification provided in the Additional Information 3B and in Figure 7. Although the RF in Additional Information 3B includes extra clusters, I wonder how come barely none of the cells in any of the muscle clusters described in *S. mansoni* are assigned to muscle progenitors nor to the muscle body cluster and yet this correspondence is very clear in Figure 7.

The ability for the RF classifier to match cell populations between schistosomes and planaria is highly dependent on how distinct the populations are within the training set (planaria). When all the labels were used, the self prediction of the planaria dataset showed little discrimination between the body muscle and muscle pharynx. In addition, there are similarities between neoblasts and most other populations in the dataset, including the muscle cells. This spreads the “muscle” signal across several classes, making it hard for the

RF classifier to predict an unambiguous match in schistosoma. This type of class-label noise makes drawing conclusions related to these populations unreliable.

In the manuscript, we emphasise that the strongest similarities between both datasets corresponds to attributes shared by the *Sm-kk7* cells of schistosomes and the *otoferlin 1* cells of planaria. The self prediction of the RF model showed no overlap between *otoferlin 1* cells and other populations. As such, these predictions will not be affected by the same class-label noise that makes predicting muscle populations challenging. Furthermore, we find that the similarities between *Sm-kk7* and the *otoferlin 1* populations are predicted confidently, regardless of number of labels used.

- 7) In line 298, the authors state “these results suggest that despite great differences in developmental stages between larval schistosomula and the asexual adult *Schmidtea mediterranea* used for this comparison, marker genes for stem cells and neuronal populations have been conserved (Figure 7)”. However, the authors use for the RF a set of 692 Variable Genes conserved between *S. mediterranea* and *S. mansoni* of unknown identity. Although amount these genes there may be some known markers of neuronal and stem cell populations, this is actually not known. Thus, the authors should provide the list of conserved variable genes used for this analysis. Additionally, to make such a statement, the authors should provide a comparison of the marker genes across cell populations in *S. mansoni* and *S. mediterranea* to show that actually some of the marker genes are conserved across species.

Following the reviewer’s suggestion, we have now provided a list of the genes used for the Random Forest (RF) analysis (Supplementary Table 5). Although the RF uses combinations of genes to make predictions, not just top marker genes, we also provide a list of manually curated marker genes that can serve as candidate genes for future studies (Supplementary Table 6).

Minor comments:

- 8) In Supplementary Figure 4e and f it is not clear what the dashed boxes mark, as the zoomed FISH images on the right do not correspond to the left pictures. If these are single confocal images, this should be stated in the Figure caption.

We included the original image of double FISH with *wnt-2* and *actin-2* in Supplementary Figure 4e. The zoomed in areas inside the dashed box in Supplementary Figures 4e and 4f do in fact correspond to the left pictures. The slight apparent discrepancy between the whole worm images and the zoomed-in images is due to the former being a maximum intensity projection of multiple z-planes while the latter is a single confocal z-section. In addition, the discrepancy between whole worm *actin-2* patterns in Supplementary Figure 4e and 4f is due to the differences between low and high *actin-2* expression within each worm - the image in Supplementary Figure 4e was taken at a lower laser power (less exposure) to avoid oversaturation of the high *actin-2* expressing cells.

- 9) In Line 193, the authors should specify that oesophageal gland cells are similar to tegument 2 cells.

We have changed this as suggested

- 10) The sentence “In addition, parenchymal cells did not co-express other cell type markers except for *actin-2*, which showed slight overlap in expression (Supplementary Figures 7F-7J)” (line 206) it is ambiguous, as it is not clear if the authors refer among parenchymal cell types or between parenchymal cell types and other cell types.

As requested, we have tried to make the text less ambiguous:

“In addition, parenchymal cells did not co-express other cell type markers that characterise other cell types, except for *actin-2* (*actin-2* muscle), which showed slight overlap in expression (Supplementary Fig. 7f-j).”

- 11) Line 370. Authors should reference Figure S11.

In our previous revision, a reference to Figure 7 was erroneously including with the following text:

“We necessarily focussed on gene expression changes amongst protein coding genes because these are now well annotated on the reference genome and can be quantified.”

However, this reference is not relevant to support the statement about annotation quality for protein coding genes, so the reference to Figure 7 has been removed and the suggestion about Figure S11, which relates to Fig 7, is no longer relevant.

Reviewers' Comments:

Reviewer #3:

Remarks to the Author:

In this revised version of the manuscript, the authors have improved significantly the quality and clarity of the manuscript. They also have clearly addressed all my previous criticisms and suggestions. I have no further comments.